# Rifampicin can induce antibiotic tolerance in mycobacteria via paradoxical changes in rpoB transcription

Jun-Hao Zhu[1,2], Bi-Wei Wang[1], Miaomiao Pan[1], Yu-Na Zeng[1], Hesper Rego[3] & Babak Javid [1]

Metrics commonly used to describe antibiotic efficacy rely on measurements performed on bacterial populations. However, certain cells in a bacterial population can continue to grow and divide, even at antibiotic concentrations that kill the majority of cells, in a phenomenon known as antibiotic tolerance. Here, we describe a form of semi-heritable tolerance to the key anti-mycobacterial agent rifampicin, which is known to inhibit transcription by targeting the β subunit of the RNA polymerase (RpoB). We show that rifampicin exposure results in *rpoB* upregulation in a sub-population of cells, followed by growth. More specifically, rifampicin preferentially inhibits one of the two *rpoB* promoters (promoter I), allowing increased *rpoB* expression from a second promoter (promoter II), and thus triggering growth. Disruption of promoter architecture leads to differences in rifampicin susceptibility of the population, confirming the contribution of rifampicin-induced *rpoB* expression to tolerance.

[1] Centre for Global Health and Infectious Diseases, Collaborative Innovation Centre for the Diagnosis and Treatment of Infectious Diseases, Tsinghua University School of Medicine, Beijing 100084, China. [2] School of Life Sciences, Peking University, Beijing 100871, China. [3] Microbial Pathogenesis, Yale School of Medicine, New Haven 06519 CT, USA. Correspondence and requests for materials should be addressed to B.J. (email: bjavid@gmail.com)

Antibiotic tolerance describes a sub-population of bacteria that are not killed, or killed more slowly than the bulk population[1–3]. Tolerance encompasses a spectrum of phenotypes[4]. At one end of the spectrum, non-dividing persister sub-populations are highly tolerant to multiple antibiotic stresses[1,5–9]. More unusually, phenotypic resistance involves not only survival, but growth in the presence of concentrations of antibiotic lethal to the bulk population[4,10–15]. Certain mutations cause increased tolerance, as opposed to resistance, to antibiotic— usually by increasing the size of the tolerant sub-population[8,14,16–18] and eradication of these hypertolerant strains may require prolonged treatment. However, the prolonged duration of antibiotics needed to eradicate certain infections including tuberculosis, are probably due to antibiotic tolerance triggered by environmental stressors or metabolic changes to a wild-type population[9–12,14,19–23].

Rifampicin is the most important first-line antimicrobial used in the treatment of drug-sensitive tuberculosis (TB), which still represents >90% of the global burden of disease[24]. The addition of rifampicin, coupled with pyrazinamide, allowed the shortening of the standard TB regimen to the current 6-month course from 18 months[25]. Resistance to rifampicin is almost entirely due to mutations in the gene coding for its drug target—the β subunit of RNA polymerase (rpoB)[26]. However, recent observations implicate multiple mechanisms for antibiotic tolerance to rifampicin[9–12,14,19,20,22,23]. Many of these mechanisms appear to be downstream of the initial trigger that promotes bacterial survival.

Here, we investigated the mechanisms governing a type of mycobacterial antibiotic tolerance specifically provoked by exposure to inhibitors of RNA polymerase (RNAP). Rifampicin exposure allowed sub-populations of mycobacteria to survive rifampicin stress and grow sufficiently to divide, which we have termed RNA polymerase-specific phenotypic resistance (RSPR). We found that RSPR is triggered by a transient increased expression of RpoB, which in turn is due to low-dose rifampicin having divergent effects on the two rpoB-rpoC promoters. Under normal conditions, expression from Promoter I inhibits maximal expression from Promoter II. Rifampicin preferentially inhibits Promoter I allowing maximal rpoB expression, mycobacterial survival, and growth.

## Results

### Rifampicin-tolerant mycobacteria grow in lethal antibiotic concentrations.
Most studies of antibiotic tolerance have focused on a non-replicating physiological state termed "persisters", in which tolerant sub-populations of cells do not grow or divide during antibiotic treatment. We sought to characterize myco-bacterial sub-populations that were tolerant to antibiotics, but actively growing[12–15]. We developed a fluorescence-based assay that would specifically identify bacteria that not only survived drug treatment, but also grew in lethal concentrations of anti-biotic (Fig. 1a). We covalently labeled the cell wall of Mycobacterium smegmatis (Msm) with a fluorescent dye, Alexa Fluor-488[10]. Due to the insertion of new cell wall material at the poles in mycobacteria, if a bacterium is able to grow in the presence of drug, as new cell wall is synthesized, the poles will become unlabeled (Supplementary Fig. 1). If the cell divides, each daughter cell will have approximately half the total fluorescence of the mother cell, and as further divisions occur, fluorescence will gradually be diluted in the population (Fig. 1a). Fluorescence from labeled M. smegmatis grown in non-selective medium was almost undetectable by flow cytometry after 16 h, which represents approximately six generation times. We then measured the fraction of dim cells (i.e., cells that had grown and divided) by flow cytometry (Fig. 1a, and Supplementary Fig. 2). The number

of bacteria that grew in rifampicin was inversely proportional to the drug concentration (Fig. 1b). This observation was confined to rifampicin—Msm cultured in streptomycin (Fig. 1b), isoniazid, kanamycin, or chloramphenicol (Supplementary Fig. 3), were unable to grow when the medium contained concentrations of antibiotic above that required to inhibit growth, the minimum inhibitory concentration (MIC). Furthermore, rifampicin treatment caused the apparently homogenous bacterial population to diverge into at least two distinct sub-groups: one that underwent active growth in the presence of bulk-lethal concentrations of drug, and the remaining bacteria (the majority at most concentrations tested), which either were killed or did not grow (Fig. 1c).

The fluorescence dilution assay allowed detection of phenotypically resistant growers over a few generations. Plating of wild-type Mycobacterium smegmatis or Mycobacterium tuberculosis (Mtb) on rifampicin-agar also showed a significant sub-population of surviving colonies, again in inverse proportion to the plated antibiotic concentration—up to 10 times the plating MIC ($MIC_{90}$)—Fig. 1d, e and Supplementary Fig. 4a. These colonies arose at significantly higher frequencies than could be explained by pre-existing rifampicin resistance-causing mutations[27]. Sequencing the rifampicin resistance determining region (RRDR) of the rpoB gene (the target of rifampicin, and the site of genetic resistance-causing mutations) from M. smegmatis colonies isolated from both low (25 μg ml[−1]) and high (100 μg ml[−1]) rifampicin-agar revealed 100%—24/24 and 6/6, respectively—to be wild-type in sequence, suggesting that the colonies were not genetically resistant to the drug[11,14]. These observations were in contradistinction to plating of Escherichia coli onto rifampicin-agar. Below the MIC, all plated bacteria were able to form colonies, and at $1 \times MIC_{90}$, 10% of plated bacteria survived and grew, as expected. With a fractional increase in drug concentration ($1.2 \times MIC_{90}$), however, the number of survivor colonies dramatically fell and were undetectable at $1.4 \times MIC_{90}$ (Supplementary Fig. 4b).

When colonies of M. tuberculosis or M. smegmatis surviving on rifampicin-agar were picked and re-plated on rifampicin-agar, there was a 10-fold increase in survival frequency (Fig. 1f, g). Furthermore, plating of rifampicin-sensitive M. tuberculosis freshly isolated from patient sputum immediately prior to starting treatment, and 1 and 3 weeks following initiation of standard therapy (see Methods) recapitulated the same phenomenon of increasing proportion of colonies on rifampicin-agar (Fig. 1h), confirming this to be a potentially clinically relevant phenotype. These observations could be consistent with "adaptive resistance"[28], which might suggest the phenomenon was mediated by mutations outside of the RRDR of rpoB. However, culturing bacteria in non-selective medium for 16 h prior to plating onto rifampicin-agar led to a complete loss of the increased phenotypic resistance, suggesting that the "adaptive phenotypic resistance" was semi-heritable, and not mediated by genetic mutation (Fig. 1g).

Our observations showed that rifampicin exposure to myco-bacteria in culture caused increased phenotypic resistance. Were the growing sub-population entirely resistant to antibiotic, or was there dynamic equilibrium of growth and death[15]? We grew AF488-stained M. smegmatis in axenic culture for 18 h in 10 μg ml[−1] rifampicin. Cells were analyzed by flow cytometry and sorted into four fractions according to fluorescence intensity (Fig. 2a) before being plated onto antibiotic-free agar plates and proportion of cells able to form colonies determined (Fig. 2b). Cells grown in the absence of rifampicin, and in 100 μg ml[−1] rifampicin were also treated in a similar manner. Close to 100% of cells grown without antibiotic formed colonies, indicating that cell-sorting itself did not adversely affect plating efficiency.

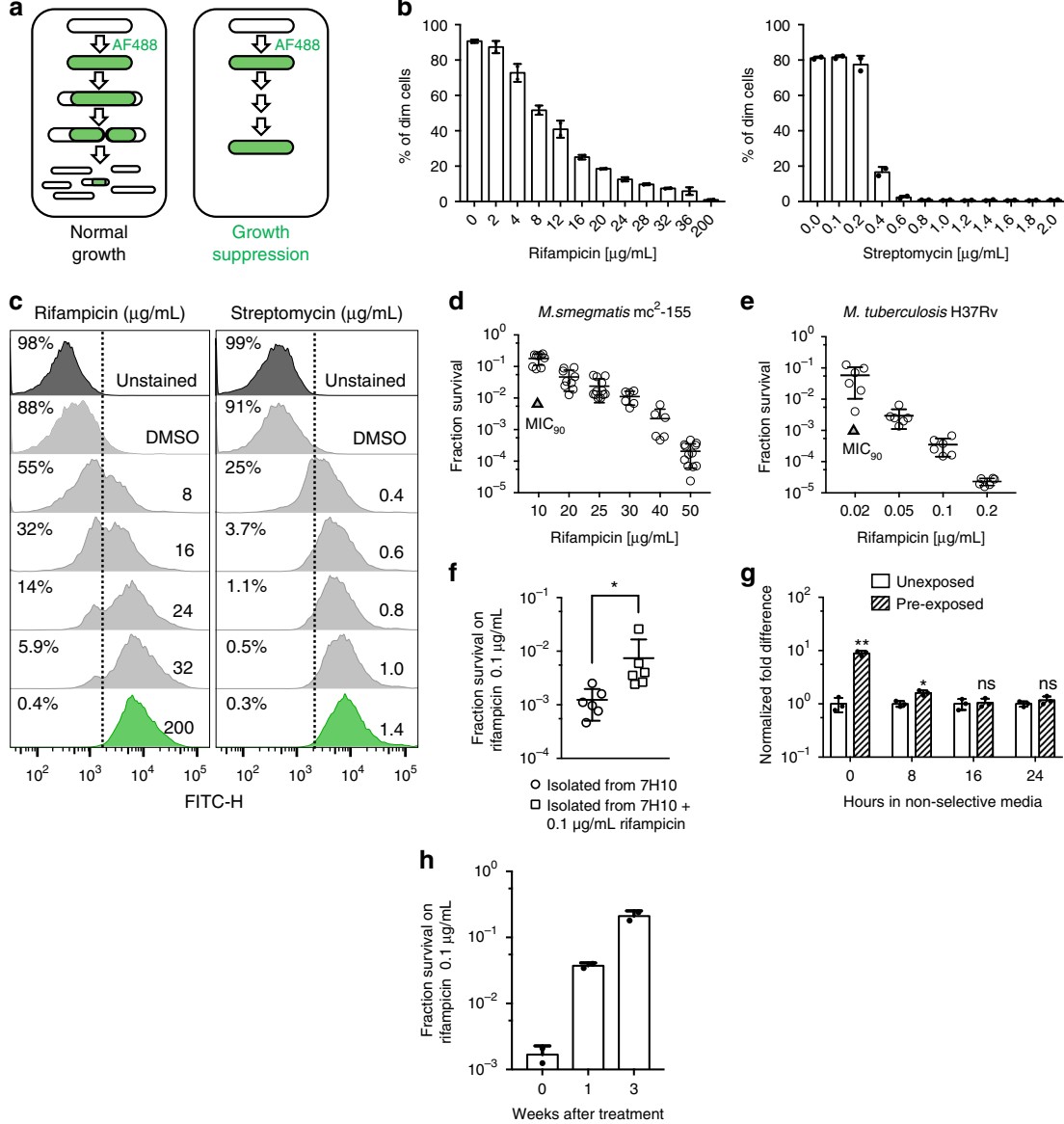

**Fig. 1** Rifampicin-tolerant mycobacteria grow in bulk-lethal concentrations of antibiotic in a concentration-dependent manner. **a** Schematic outlining the basis for the fluorescence dilution assay: cells are stained with Alexa-fluor-488 (AF488), and as they grow and divide, the total fluorescence is diluted (left panel). If cells fail to grow—either due to death or a non-replicating physiological state, full fluorescence is retained (right panel). **b** Fluorescence dilution assay of *M. smegmatis* exposed to indicated concentrations of rifampicin (left) and streptomycin (right) and analyzed by flow cytometry (see Supplementary Fig. 2 for flow cytometry strategy for scoring dim cells). Bars represent duplicate experiments. **c** Sample flow cytometry histograms of fluorescence distributions of single *M. smegmatis* cells following 16-h exposure to rifampicin (left panel) or streptomycin (right panel) at indicated concentrations. *M. smegmatis* (**d**) or *M. tuberculosis* (**e**) were plated on rifampicin-agar at varying concentrations and the fractional survival (number of colonies on rifampicin-agar compared with non-selective medium) calculated. Results represent 7–12 biological replicates per concentration. **f** Plating rifampicin tolerance of *M. tuberculosis*-H37Rv from colonies picked from non-selective medium (7H10 plates) or previously plated on rifampicin-agar. The picked colonies were resuspended and plated on rifampicin-agar as in (**e**). **g** Three colonies of *M. smegmatis* that grew on non-selective medium ("unexposed") or three that survived and grew on 25 μg ml$^{-1}$ rifampicin-agar ("pre-exposed") were picked and re-suspended in complete 7H9 medium for the indicated time without antibiotics and then plated onto 25 μg ml$^{-1}$ rifampicin-agar or non-selective medium to calculate fractional survival. Results normalized to fractional survival of unexposed colonies at time = 0. *$p < 0.05$ and **$p < 0.01$ by Student's *t*-test. **h** Plating rifampicin tolerance was determined from freshly isolated *M. tuberculosis* from sputum of a treatment naive patient immediately prior and following initiation of treatment with the standard regimen (see Methods). Results are representative of experiments performed for two distinct patients

As expected, with decreased fluorescence (indicating growth), the plating efficiency increased, with 40% of the least fluorescent cells able to form colonies, decreasing to approximately 1% of cells that retained full fluorescence (Fig. 2b). Of note, in the culture treated with high-dose (100 μg ml$^{-1}$) rifampicin, 0.1% of cells were still able to form colonies, and presumably represented classical persisters[1]. Furthermore, our data indicated that phenotypically

resistant growers, or their progeny, were not guaranteed to survive rifampicin killing and that growers, persisters and killed cells co-existed upon rifampicin treatment.

In classical antibiotic tolerance, exposure of the bulk bacterial population to lethal concentrations of drug kills the more susceptible sub-population, leaving the more tolerant sub-population alive[2]. Was the increase in bulk phenotypic resistance

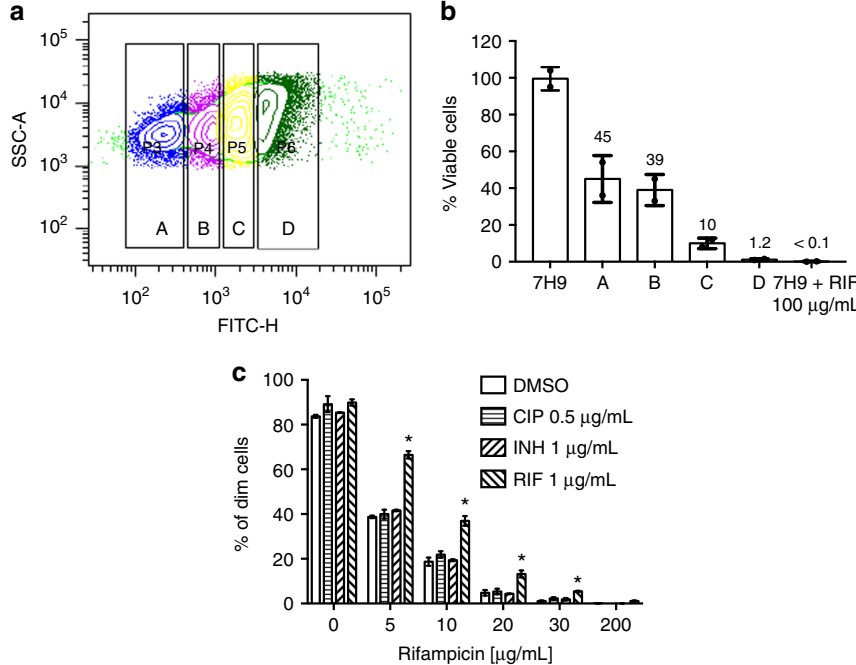

**Fig. 2** *Mycobacterium smegmatis* exposed to rifampicin results in a distinct and specific rifampicin tolerance. **a** Gating strategy for sorting of *M. smegmatis* by fluorescence intensity following 18-h exposure to 10 µg ml$^{-1}$ rifampicin in axenic culture. In all, 10$^6$ cells were sorted from each gate, and then 1000 bacteria from each sub-population plated onto drug-free agar medium. **b** Indicated sorted sub-populations from (**a**) and *M. smegmatis* exposed to 100 µg ml$^{-1}$ rifampicin and drug-free medium were then plated on non-selective medium to calculate survival. Error bars represent standard deviation of two technical replicates. **c** *M. smegmatis* was exposed to sub-MIC concentrations of the indicated antibiotics or vehicle for 3 h, then washed, stained with AF488 and then inoculated into 7H9-rifampicin at the indicated concentrations. After 16-h incubation, fluorescence dilution was determined by flow cytometry as before. Error bars represent biological duplicates. *$p < 0.05$ by Student's *t*-test

witnessed upon rifampicin exposure (Figs. 1f–h) due solely to selection of the more tolerant sub-population? We cultured Msm ± rifampicin in axenic culture. At indicated times, aliquots were then spread onto agar medium ± rifampicin. In the absence of rifampicin, the proportion of cells able to survive and grow on rifampicin-agar was stable (Supplementary Fig. 5). With rifampicin treatment (4 × MIC), the proportion of phenotypically resistant cells increased dramatically, suggesting that selection of a tolerant sub-population was occurring (Supplementary Fig. 5). However, treatment with 1 µg ml$^{-1}$ rifampicin, a concentration well below the MIC, and insufficient to kill the vast majority of cells, also resulted in a significant increase in the proportion of phenotypically resistant bacteria (Fig. 2c), suggesting that rifampicin triggered an adaptive programme that increased survival to the drug. This effect was both specific for rifampicin: exposure to sub-MIC concentrations of other antibiotics did not increase rifampicin phenotypic resistance (Fig. 2c), and dominant: exposure to other antibiotics did not mask the phenotype (Fig. 1h) suggesting that the survival programme was not a generalized response to environmental stress. Furthermore, rifampicin exposure specifically increased tolerance to rifampicin, not other antibiotics (Supplementary Fig. 6), suggesting in turn that rifampicin did not induce a generalized survival response. To determine whether the response was specific to rifampicin or to agents targeting RNA polymerase more broadly, we tested fidaxomicin, an inhibitor of the RNAP "switch region"[29] in the fluorescence dilution assay. Pre-treatment of *M. smegmatis* with fidaxomicin but not streptomycin (targeting translation) resulted in increased phenotypic resistance to rifampicin (Supplementary Fig. 6e), verifying that the phenotype was mediated by inhibitors of RNAP. We therefore decided to name the tolerance induced by rifampicin (or fidaxomicin) exposure as RSPR.

How does RSPR relate to mistranslation-mediated rifampicin tolerance[11,14]? We compared wild-type *M. smegmatis* and a strain we previously characterized (HWS.4) with a mutation in *gatA* conferring high rates of translational error in the indirect transfer RNA aminoacylation pathway and associated rifampicin tolerance[14] in the fluorescence dilution assay in the presence of bulk-lethal concentrations of rifampicin. As expected, the high mistranslating strain had a larger proportion of 'growers' compared with wild-type (Supplementary Fig. 6f). The proportion of growers increased further with sub-MIC rifampicin pre-treatment, potentially explaining the very high rates of rifampicin tolerance observed in this strain[14].

**RSPR is semi-heritable and correlates with RpoB accumulation.**
Since our data suggested inhibition of RNAP activity was required to trigger RSPR, we decided to measure the expression of RpoB, the target of rifampicin, in single *M. smegmatis* cells. Variation in abundance in an antibiotic target can have divergent effects on drug susceptibility[30,31]. To measure the abundance of the target of rifampicin, RpoB, in *M. smegmatis* exposed to rifampicin we constructed strains by recombineering where the native *rpoB* gene was tagged C-terminally with one of two fluorescent proteins (FPs; Supplementary Fig. 7a). The intracellular distribution of RpoB-FP resembled the nucleoid, as expected (Supplementary Fig. 7b). Exposure of cells to sub-MIC (1 µg ml$^{-1}$) rifampicin, but not other antibiotics led to a significant induction of RpoB-mEmerald within 3 h as measured by both microscopy and flow cytometry (Fig. 3a, b and Supplementary Fig. 8). Using time-lapse microscopy, we measured growth and fluorescence in cells that survived treatment with a concentration (20 µg ml$^{-1}$) of rifampicin that is lethal to the bulk population. Expression of RpoB-mApple was stable in the absence of antibiotic, but accumulated

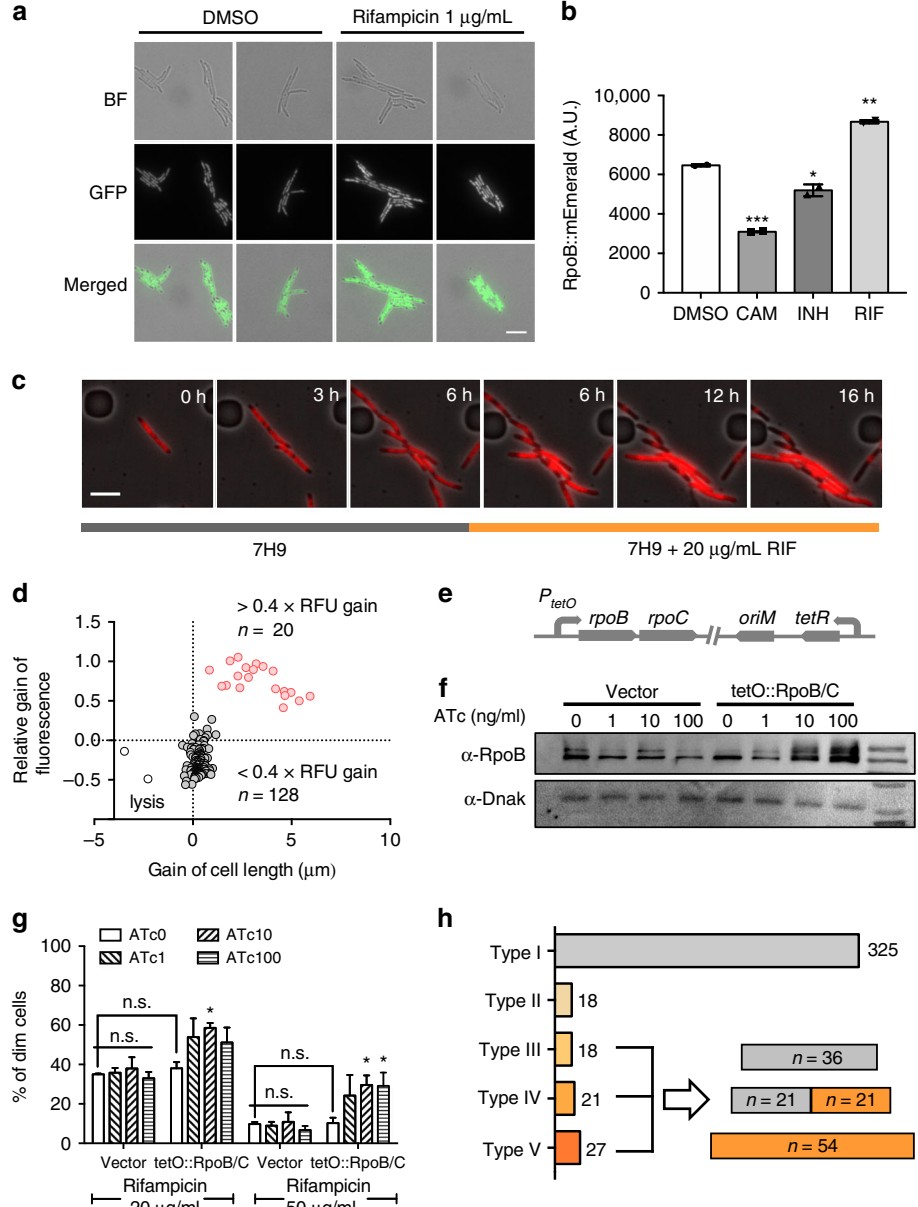

**Fig. 3** RSPR is semi-heritable and correlates with the accumulation of RpoB, the target of rifampicin. **a** Fluorescence microscopy of *M. smegmatis rpoB-mEmerald* after exposure to sub-MIC rifampicin or vehicle for 3 h showing accumulation of fluorescent signal after rifampicin. **b** Flow cytometric analysis of green fluorescence (representing RpoB-mEmerald) after 3-h exposure to sub-MIC (non-bulk lethal) concentrations of indicated antibiotics or vehicle. Data represent biological duplicates as measured by flow cytometry. *$p < 0.05$, **$p < 0.01$ and ***$p < 0.001$ by Student's $t$-test. **c** Fluorescence microscopy of representative image series of a microcolony visualized in a microfluidic chamber following growth in 7H9 and then 7H9-rifampicin. **d** Change in cell length and fluorescent intensity (RpoB-mApple) of 150 cells following addition of 20 μg ml$^{-1}$ rifampicin to the flow chamber. Cells were scored following 6 h of treatment. Red circles represent *growers*, gray circles cells that did not grow and the two cells that underwent lysis are represented by hollow circles. **e** Schematic of the construct for *in trans* overexpression of *rpoB-rpoC* using a tetracycline-inducible promoter. **f** Western blot for RpoB following induction of expression with anhydrotetracycline (ATc). **g** Fluorescence dilution assay in rifampicin at indicated concentrations for *M. smegmatis* overexpressing *rpoB-rpoC* from the construct in (**e**) or vector only control. **h** Categorization of 409 cells and the progeny of 66 dividing cells into the five RSPR response types (Supplementary Fig. 13) following treatment with 20 μg ml$^{-1}$ (bulk-lethal concentration) rifampicin for 16 h

rapidly upon rifampicin exposure in RSPR cells (Fig. 3c, Supplementary Fig. 9a, b, 10 and Supplementary movie 1). Following one cell and its progeny through three divisions prior to subsequent exposure to rifampicin, in 4/8 cells, fluorescence dimmed with apparent nucleoid contraction, consistent with cell death upon rifampicin treatment. The other four cells all exhibited strong accumulation of intracellular fluorescence and subsequent cell elongation (Supplementary Fig. 10 and Supplementary movie 1). The fluorescence intensity and cell elongation rate in

two siblings with differing fates upon rifampicin exposure are illustrated in Supplementary Fig. 10b, c: there was a clear correlation between accumulation of RpoB-mApple and cell elongation. Measuring both parameters in 150 cells over 6 h of rifampicin treatment identified three distinct populations. The majority of cells exhibited decreased fluorescence and invariant cell length. Approximately 13% of cells grew, associated with increased RpoB-mApple intensity, and two cells lysed (Fig. 3d).

RpoB-mApple accumulation occurred after rifampicin exposure, but preceded cell elongation. We determined the lag time ($\tau_{lag}$) of both RpoB-mApple induction and cell growth of 33 *grower* cells treated with 20 µg ml$^{-1}$ rifampicin (see Methods). Despite cell–cell variability in lag time for both parameters, RpoB-mApple accumulation preceded re-initiation of growth by an average of 1.5 h (Supplementary Fig. 11a). When cells were treated with varying concentrations of rifampicin, lag time was positively correlated with rifampicin concentration—i.e., higher drug concentrations led to longer lag times (Supplementary Fig. 11b). Taken together, these data suggest that RpoB accumulation following rifampicin exposure is a physiological hallmark for the growth-responsive RSPR sub-population.

Increased expression of RpoB was associated with survival and growth in rifampicin. Was the survival due to increased number of target (RpoB) molecules for a given amount of antibiotic[31]? If so, we would predict that RpoB abundance prior to rifampicin exposure would also be correlated with survival and growth, however, no such association was identified (Supplementary Fig. 12), suggesting that RpoB accumulation upon rifampicin exposure leads to a specific survival programme, possibly through the specific transcription of late-adaptive genes. Nonetheless, artificial overexpression of *rpoB-rpoC* via a tetracycline-inducible promoter expressed *in trans* (Fig. 3e, f) to levels higher than typically seen with rifampicin exposure did lead to increased survival to high-dose rifampicin (Fig. 3g), suggesting that supraphysiological and sufficiently high expression of RNAP may also contribute to tolerance.

Exposure to bulk-lethal rifampicin concentrations segregates the Msm bulk population to several differing responses. We categorized these responses as observed by microscopy of micro-colonies in microfluidic devices into five different types (Supplementary Fig. 13): most cells did not overexpress RpoB and stopped growing (Type I). The four remaining categories all accumulated RpoB, with diverging fates. Type II cells started cellular elongation but growth arrested prior to division. Types III–V all grew sufficiently to divide. Both daughter cells of Type III cells immediately stopped growing following division. In Type IV cells, one daughter resumed growth, whereas the other daughter underwent growth arrest, and in Type V cells, both daughters resumed growth following division. No cells grew without prior accumulation of RpoB. We determined the cell types of 409 cells. Approximately 80% of cells did not accumulate RpoB upon rifampicin exposure. Of the remaining 20% (84/409), 66 cells (16% of total) grew sufficiently to divide into 132 daughter cells (Types III–V). Of these second-generation cells, over 55% were capable of re-starting growth (Fig. 3h). These data support the hypothesis that semi-heritable RSPR (Fig. 1g) is due to both rifampicin-mediated selection and semi-heritable rifampicin-induced adaptation. Variations in cell size and differential RpoB segregation between siblings were measured but were unable to explain the difference in survival between Type IV siblings or Type IV and V cells (Supplementary Fig. 14).

**Rifampicin upregulates *rpoB* via differential effects on two promoters**. We wished to determine whether rifampicin-induced RpoB expression was transcriptionally or translationally regulated. First, we examined the promoter sequence of *rpoB*. In keeping with many bacteria, β (RpoB) and β′ (RpoC) are expressed from a single transcript. In *E. coli*, *rpoB* and *rpoC* share a promoter with two upstream ribosomal genes[32], however, in mycobacteria, *rpoB-rpoC* are the sole genes in the operon[33,34]. We compared the sequence upstream of the coding region for *rpoB* among three mycobacterial species, and identified three highly conserved non-coding regions: two SigA-dependent promoters that had been

previously annotated[33,34] and a short inter-promoter region (Fig. 4a). The –35 region of the distal (5′) promoter (Promoter I) was approximately 330-bp upstream of the start codon for *rpoB*. To retain the complete transcriptional regulatory elements of the operon, we fused the sequence 500-bp upstream of the coding segment of Msm *rpoB-rpoC* with the coding sequence for the green FP mEmerald—P$_{rpoBC}$-mEmerald—as a surrogate to measure transcriptional expression from the native promoter. We transformed this construct into the *M. smegmatis* strain expressing RpoB-mApple on the chromosome at the native site, and measured green (mEmerald, from the promoter reporter) and red (mApple, from the protein fusion reporter) fluorescence in response to rifampicin treatment. In the absence of drug, expression from both constructs was fairly homogenous throughout the population, but with antibiotic exposure, both red and green fluorescence increased significantly in a sub-population of cells, whereas in the other, susceptible sub-population, expression of both FPs had dimmed considerably (Fig. 4b, Supplementary Fig. 15a–c and Supplementary movie 2). Expression from the promoter was upregulated in response to sub-MIC concentrations of rifampicin, but not other antibiotics (Supplementary Fig. 16a, Supplementary Fig. 6g). The response was also specific to the *rpoB-rpoC* promoter: expression of mEmerald, fused to two commonly used mycobacterial promoters P$_{hsp60}$ and P$_{smyc}$[26,35] did not result in increased expression following rifampicin exposure (Supplementary Fig. 15d).

The second (3′) promoter in the *rpoB-rpoC* operon (Promoter II) has a conserved 5′-CGCTATNGTT-3′ motif that has been annotated as driving strong transcriptional initiation[33], whereas the first (5′) promoter (Promoter I) has been classified as contributing the minority of transcripts from the operon[33]. To determine the relative contribution of the two promoters in the operon toward rifampicin-induced RpoB expression, we quantified the relative abundance of the two different 5′ UTRs in both *M. smegmatis* and *M. bovis*-BCG (Bacillus Calmette Guérin, BCG—which has an identical promoter structure to *M. tuberculosis*) following sub-MIC rifampicin exposure (Supplementary Fig. 16a–c). The relatively weak expression from the first Msm (5′) promoter (Promoter I) was further downregulated by 30% with rifampicin treatment compared with the no-treatment control. By contrast, expression from Promoter II and transcripts encompassing the coding region of mRNA were significantly upregulated after rifampicin exposure (Fig. 4c). As with RSPR, upregulation of *rpoB* expression was due to RNAP inhibition, since sub-MIC fidaxomicin but not other antibiotics acted similarly to rifampicin (Supplementary Fig. 16a), and rifampicin acted in a dominant fashion: exposure to both rifampicin and isoniazid still caused upregulation of *rpoB* (Supplementary Fig. 16d).

The differentially regulated transcriptional activity from Promoters I and II suggest they may play distinct roles in rifampicin-induced *rpoB* expression. To test this, we made a number of truncated promoter constructs to drive mEmerald expression (Supplementary Fig. 17) in *M. smegmatis*. Exposure to sub-MIC rifampicin caused 1.6-fold increased expression of mEmerald in constructs harboring the intact annotated *rpoB-rpoC* promoter. In the absence of Promoter I, overall expression in the absence of rifampicin was enhanced 1.6-fold, but rifampicin exposure no longer led to increased expression. Further truncations, also disrupting Promoter II abrogated expression and verified that there were no occult promoters regulating expression between Promoter II and the start codon of *rpoB* (Fig. 4d). Similar experiments in BCG showed broadly the same phenomenon (Supplementary Fig. 18) with the exception that, as long as Promoter II was intact, rifampicin exposure consistently upregulated mEmerald.

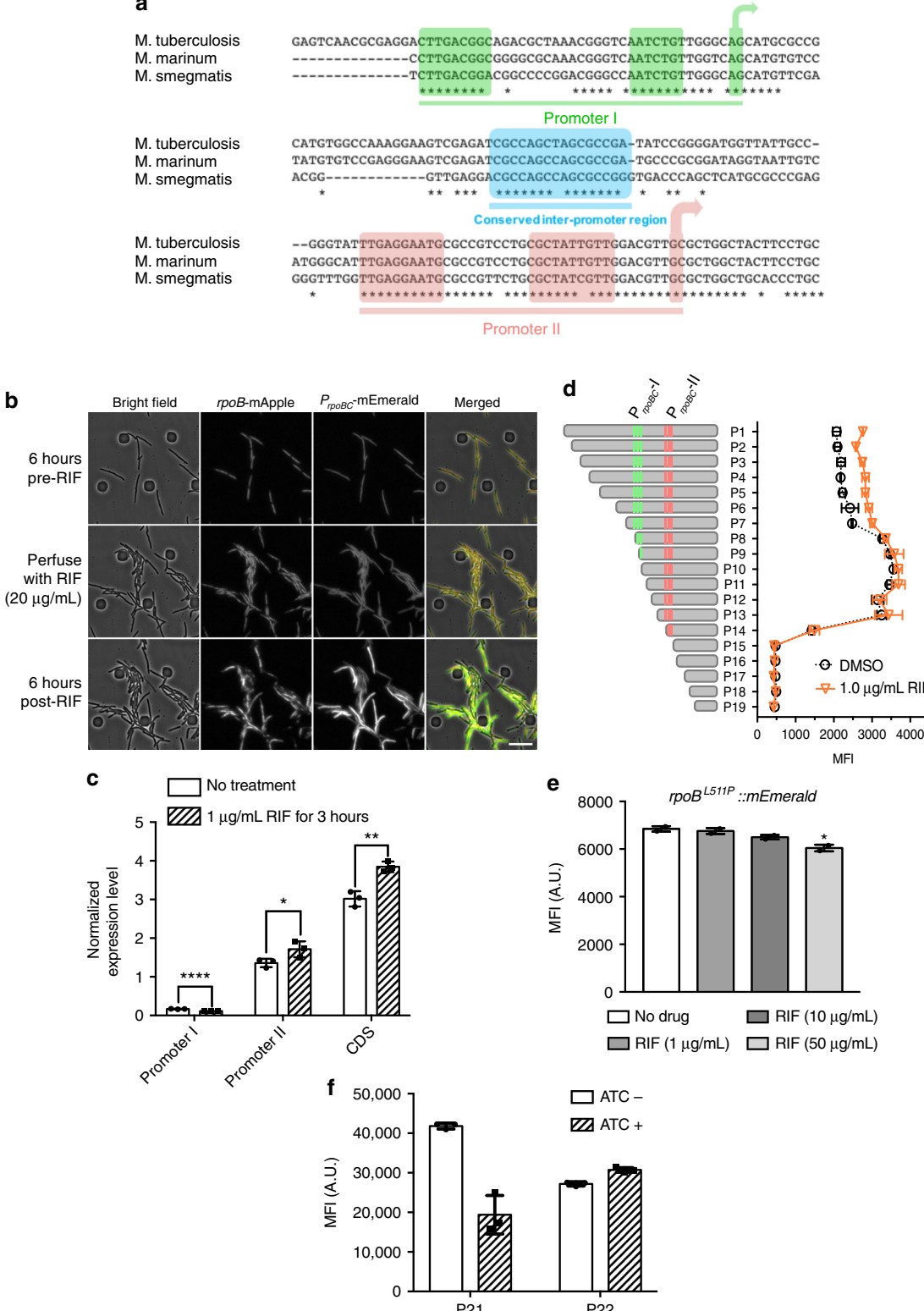

We also constructed a synthetic promoter variant with intact Promoter I, but lacking Promoter II (P20, Supplementary Fig. 19a). In keeping with Fig. 4c, the net contribution of Promoter I was 8% of the intact dual promoter, and was significantly inhibited in the presence of 1 μg ml$^{-1}$ rifampicin to basal detection levels, suggesting that the two promoters individually have divergent responses to rifampicin.

Our data suggested: (a) in Msm, Promoter I was required for the rifampicin-induced expression of *rpoB*, (b) expression from Promoter I was inhibited by low-dose rifampicin in both Msm and BCG and (c) inhibition of Promoter I by low-dose rifampicin led to increased expression from Promoter II. Based on these observations, we hypothesized that *in the absence of rifampicin*, expression from Promoter I suppressed expression

**Fig. 4** Rifampicin exposure upregulates *rpoB* by differential susceptibility of its two promoters to inhibition by rifampicin. **a** Sequence alignment of the conserved mycobacterial *rpoB-rpoC* operon for *M. tuberculosis*-H37Rv, *M. marinum* and *M. smegmatis* mc$^2$-155. The operon is controlled by two promoters, the 5′ Promoter I and the 3′ Promoter II (−35 and −10 elements boxed), with a conserved inter-promoter region (shaded blue). The two transcription start sites (TSSI and TSSII) are also illustrated. **b** Representative fluorescence microscopy images of RpoB-mApple and mEmerald, driven by the *rpoB-rpoC* promoter (P$_{rpoBC}$-mEmerald) before, immediately following and 6 h after exposure to rifampicin in the flow chamber. **c** Relative mRNA abundance (normalized to *sigA* mRNA) of Promoter I, Promoter I + II (Promoter II) and coding sequence (CDS) transcripts from rifampicin (1 µg ml$^{-1}$—for 3 h) or vehicle-treated *M. smegmatis*—see Supplementary Fig. 17 for promoter-specific primers. Each bar represents four biological replicates. *$p < 0.05$, **$p < 0.01$, ****$p < 0.0001$ by Student's *t*-test. **d** Mean fluorescent intensity (MFI) of mEmerald driven by 19 *rpoB-rpoC* promoter truncations (see Supplementary Fig. 17b) as measured by flow cytometry in response to sub-MIC rifampicin for 3 h or vehicle. **e** MFI of RpoB$^{L511P}$-mEmerald in response to sub-MIC concentrations of rifampicin or vehicle for 3 h. Each bar represents biological duplicates. *$p < 0.05$ by Student's *t*-test. **f** MFI of mEmerald expression driven by the two chimeric promoters—see Supplementary Fig. 20d, e ±50 ng ml$^{-1}$ ATc for 6 h measured by flow cytometry in *M. smegmatis* strains expressing the promoter constructs. Each bar represents three biological replicates.

from Promoter II, leading to sub-maximal expression. Upon even low-dose rifampicin exposure, expression from Promoter I was preferentially inhibited, relieving its inhibitory effect on Promoter II, leading to increased overall expression of *rpoB*. To validate this model, we first tested the in vitro sensitivity of Promoters I and II, individually and together, as well as a control promoter P*smyc* to inhibition of transcription by rifampicin. As can be seen in Supplementary Fig. 19b, mRNA synthesis from Promoter I was more susceptible to rifampicin inhibition than the other promoters, indicating that the hypersusceptibility to rifampicin of this promoter was not due to factors *in trans*, but likely due to the intrinsic promoter structure itself. The differential susceptibility was relative: higher concentrations of rifampicin inhibited transcription from all promoters (Supplementary Fig. 19b). We also made a C-terminally tagged RpoB-mEmerald construct on an Msm rifampicin-resistant (RpoB-L511P) background. As predicted, this strain was insensitive to rifampicin with regard to upregulation of RpoB expression (Fig. 4e), further validating the requirement of rifampicin-sensitive RNAP for rifampicin-induced RpoB upregulation.

To determine the dependence on rifampicin-sensitive RNAP for the observed phenotype, we made an Msm strain expressing rifampicin-resistant RpoB$^{L511P}$-mEmerald, as above, but also expressing wild-type, but tetracycline-regulated expression of *rpoB in trans*. In the absence of anhydrotetracycline (ATc), the expression of RpoB-mEmerald remained insensitive to rifampicin, as before. However, titration of rifampicin-sensitive RpoB by ATc led to a dose-dependent rifampicin-responsive induction of RpoB-mEmerald (Supplementary Fig. 19c), suggesting that inhibition of RNAP transcription by rifampicin was necessary for rifampicin-induced RpoB expression.

Furthermore, our model predicted that if expression from Promoter I was increased, this would lead to further inhibition from Promoter II. We therefore constructed two chimeric promoters, P21 and P22 (Fig. 4f and Supplementary Fig. 19d). P21 replaced Promoter I with a tetracycline-inducible promoter P*smyc-tetO*, but retained the native inter-promoter sequence of the *rpoB-rpoC* operon. In P22, P*smyc-tetO* replaced Promoter I, and an arbitrary sequence of the same length replaced the conserved inter-promoter sequence. These promoters drove mEmerald expression, as before, and the construct was cloned into a plasmid constitutively expressing the *tet* repressor[35] and transformed into *M. smegmatis*. In the strain with P21-driven mEmerald, there was strong constitutive expression of green fluorescence, consistent with the high promoter strength of Promoter II. However, addition of ATc, which induced expression from P*smyc-tetO*, caused significant attenuation of mEmerald expression (Fig. 4f). To exclude the possibility of transcriptional interference[36], induction of a strain expressing P22, without the conserved inter-promoter region failed to attenuate green fluorescence (Fig. 4f). Therefore, in *M. smegmatis*,

Promoter I negatively regulated expression from Promoter II via transcription of the conserved, inter-promoter region.

**Disruption of *rpoB* promoter architecture alters rifampicin resistance.** Finally, our model predicted that if the native structure of the *rpoB-rpoC* promoters were disrupted, this would in turn, alter rifampicin tolerance. We constructed a *M. smegmatis* strain with merodiploid expression of *rpoB-rpoC* from the Giles phage integration site on the mycobacterial chromosome but at the second site, only Promoter II drove expression of the genes. The native chromosomal copy of the *rpoB-rpoC* operon was then deleted by homologous recombination. The Promoter II-only strain had three- to fourfold increased rifampicin tolerance, consistent with the release of inhibition of *rpoB* expression in the absence of Promoter I (Fig. 5a). Importantly, exposure of the wild-type strain to sub-inhibitory concentrations of rifampicin led to >10-fold increase in RSPR, consistent with earlier findings (Figs. 1f–h), but a significantly blunted response in the Promoter II strain (Fig. 5b), suggesting that the native promoter structure allows a tuning of rifampicin tolerance following rifampicin exposure. There was still a slight increase in rifampicin tolerance in this strain following rifampicin exposure, suggesting that either the new chromosomal location of the Promoter II-driven *rpoB-rpoC* locus, or other downstream factors may also play a role in tuning of rifampicin tolerance.

## Discussion

The extremely long duration of the standard TB regimen is probably due to antibiotic tolerance, and tolerance to rifampicin, which is the backbone of the regimen, contributes significantly to the extended length of therapy. Here, we describe both a novel and distinct form of tolerance—induced specifically by exposure of mycobacteria to rifampicin or other RNAP inhibitors, and conferring tolerance solely to rifampicin itself, which we have termed RSPR. Of note, a small sub-population of mycobacteria, inversely proportional in size to the dose of rifampicin, not only survive, but also grow in the presence of bulk-lethal concentrations of drug following a rifampicin-induced increased expression of RpoB (Fig. 6). Other studies have identified increased expression of *rpoB* following exposure to rifampicin[37–39]. We find that this is consistently true and have described its mechanism, and its role in RSPR. What is the relation between RpoB-mediated RSPR and other forms of rifampicin tolerance? We had previously shown that mistranslation of a critical residue in the RRDR of RpoB *prior to drug exposure* contributed significantly to rifampicin tolerance[14]. We now show that strains with high mistranslation due to mutations in *gatA* have increased basal RSPR, as described before, which increases further upon rifampicin exposure. The positive interaction of these two distinct mechanisms for rifampicin tolerance may explain the observation

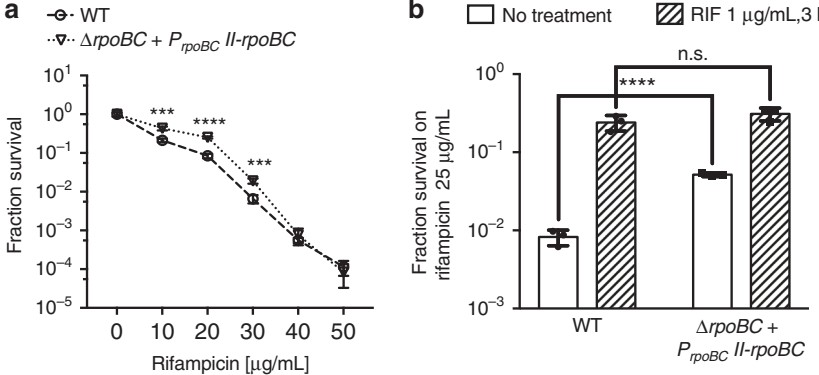

**Fig. 5** Disruption of the *rpoB* promoter architecture alters RSPR. **a** Fractional survival on 25 μg ml⁻¹ rifampicin-agar of wild-type *M. smegmatis* or *M. smegmatis* with only Promoter II-driven *rpoB-rpoC* at the Giles chromosomal location. Data represent biological triplicates. ***$p < 0.001$ and ****$p < 0.0001$ by Student's *t*-test. **b** Cultures of the two strains in (**a**) were pretreated with 7H9 ± 1 μg ml⁻¹ rifampicin for 3 h and then plated on rifampicin-agar and fractional survival calculated. Data represent biological triplicates. ****$p < 0.0001$ by Student's *t*-test, n.s. no significant difference by Student's *t*-test

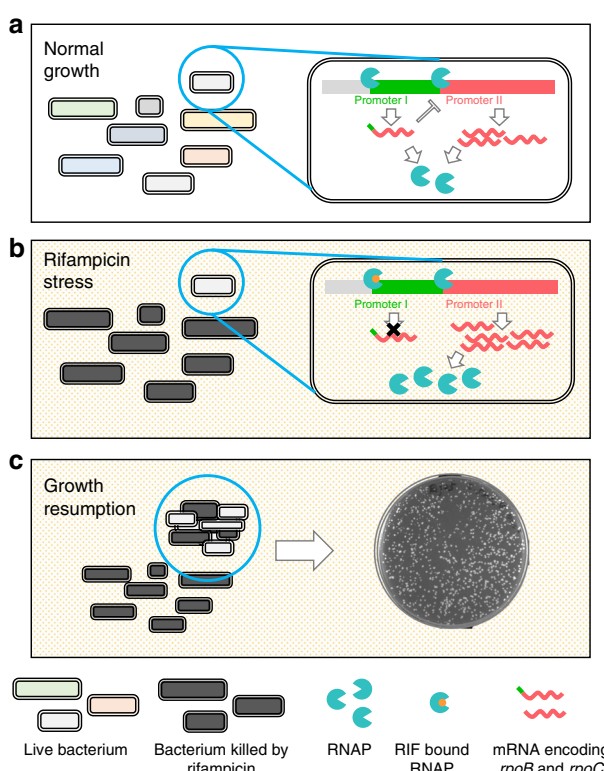

**Fig. 6** Working model of transcriptional regulation of *rpoB-rpoC* in mycobacteria and its role in rifampicin-specific phenotypic resistance. **a** Under normal growth conditions, maximal *rpoB* expression is suppressed by the interaction of the promoters in the *rpoB-rpoC* operon. Expression from Promoter I prevents maximal expression from the stronger Promoter II. **b** Upon encountering rifampicin, some cells are killed by the antibiotic. In surviving cells, expression from Promoter I is completely inhibited by minimal concentrations of rifampicin. This in turn relieves the inhibitory effect on expression from Promoter II, allowing maximal RpoB expression (in Msm, and possibly additional other mechanisms in BCG and Mtb), which then allows initiation of a specific adaptive transcriptional programme. **c** Increased expression of RNAP in response to rifampicin allows cells to survive moderate but otherwise lethal concentrations of rifampicin and after a delay, resume growth and division, with adaptive, semi-heritable hypertolerance to the drug in the face of ongoing exposure

that mistranslating mycobacterial strains have orders of magnitude increased tolerance to rifampicin[11,14]. Here, neither cellular content of RpoB prior to drug exposure, nor cell size[23] were correlated with survival and growth. Our studies examined survival and growth in the presence of rifampicin, whereas Richardson et al. measured survival following drug exposure[23], potentially explaining the differences observed. We could not detect differences in RpoB content of sister Msm cells with differing survival fates to rifampicin immediately pre-division (Supplementary Fig. 14), although these measurements were taken at a single time-point, and therefore may not have detected subsequent changes in RpoB abundance.

Only cells that survived the initial exposure to rifampicin, possibly due to having a proportion of rifampicin-resistant RNA polymerase due to mistranslation[14], and were subsequently able to upregulate *rpoB* expression, were able to grow in the presence of bactericidal concentrations of antibiotic. This suggests that rifampicin-induced RpoB expression and subsequent RSPR is predominantly an early-adaptive response to rifampicin exposure. The increased abundance of RNAP polymerase, may in turn allow expression of subsequent survival programmes, such as increased expression of antibiotic efflux pumps[19]. Heterologous expression of RpoB-RpoC via a tetracycline-inducible plasmid led to increased tolerance, even at very low (1 ng ml⁻¹ ATc) levels of induction that did not result in observable increase in total RpoB by western blot (Fig. 3f). This may be due to highly variable cell–cell variation in RpoB expression in a minor sub-population mediating the phenotype, which would not be detected in the bulk population by western blot. Although supra-physiological expression of RpoB-RpoC led to increased rifampicin tolerance (Figs. 3g, 5a), our data are more consistent with upregulation of RNAP initiating a specific survival programme. Rifampicin tolerance is likely to be a coordinated series of events, starting with initial survival to drug[11,14,23], an early-adaptive response predominated by a rifampicin-induced expression of RpoB, which in turn triggers growth, and late-adaptive responses, such as efflux pump expression[19].

Rifampicin is an inhibitor of transcription. However, exposure to rifampicin is associated with specific upregulation, as well as downregulation of gene expression in a number of bacterial species[37,40]. Innate promoter architecture was associated with at least a subset of both types of responses in *Salmonella*[40]. In *E. coli*, the *rpoB-rpoC* genes are part of an operon shared, 5′ with two ribosomal genes[32,41]. Readthrough of a transcriptional attenuator between the ribosomal genes and the genes encoding β and β′ is

responsible for the rifampicin-induced transcriptional response[41], however, in mycobacteria *rpoB-rpoC* are the only two genes in their operon[34]. We did not identify evidence of readthrough upstream of the operon in the presence or absence of rifampicin. Transcriptional interference may impede expression of adjacent promoters[36], but the inherent promoter strength of Promoter I in the mycobacterial *rpoB-rpoC* operon (Fig. 4) was too low compared with Promoter II to describe our observation, and the inter-promoter sequence was necessary for attenuation of *rpoB* expression in Msm (Fig. 4f). The regulation of *rpoB-rpoC* expression in our system is more representative of a type 4 incoherent feed[42]. Global transcriptional modulators such as NusA, CarD, and RbpA have a role in stabilizing the transcription initiation complex[43–46]. These modulators do not uniformly affect all promoters as revealed by mapping using chromatin immunoprecipitation and sequencing (ChiP-seq)[43,44], and may therefore play a role in the differential rifampicin susceptibility of the two *rpoB-rpoC* promoters and indeed, the interaction of CarD with RNAP has been implicated in mycobacterial rifampicin sensitivity[45].

A distinctive characteristic of RSPR compared with other forms of antibiotic tolerance is the clear inverse relationship between the antibiotic dose and the size of the tolerant sub-population (Fig. 1). In other forms of tolerance, such as persister bacteria, the tolerant sub-population is usually tolerant to multiple antibiotics and the size of the tolerant sub-population does not vary with drug concentration[1,2]. Recent studies measuring antibiotic concentrations in disease lesions suggests considerable variability in the dose of rifampicin that reaches the necrotic center of granulomata[47]. Our preliminary and limited investigation of RSPR in two clinical isolates suggests that it may be clinically relevant. Our model would predict that a relatively straightforward way to decrease the tolerant sub-population would be by increasing the effective antibiotic concentration[48]. Decreased rifampicin plasma concentrations are associated with poor clinical outcome in clinical tuberculosis[49,50]. Several clinical trials have examined increasing rifampicin dosing in the treatment of tuberculosis[51–53] with the suggestion of improvement of clinically relevant parameters and outcome in some[51,52] but not all[53] of these studies, suggesting that this may be an important way to overcome rifampicin-induced RSPR. Ongoing clinical trials[54–56], coupled with minimally invasive rifampicin pharmacokinetic monitoring[57] may determine whether such a strategy, which our data support, can shorten the duration of standard tuberculosis therapy.

Rifampicin is the most important antibiotic in the standard anti-tuberculous arsenal. Mycobacteria—including the model organism *M. smegmatis* and members of the *M. tuberculosis* complex have evolved a specific mechanism to tolerate rifampicin stress by upregulating its cellular target in response to exposure via a differential response to the drug's actions by the promoters regulating *rpoB* expression. Mechanistic understanding of rifampicin tolerance[58] may allow targeted therapeutics to speed up tuberculosis treatment.

## Methods

**Bacterial strains, culture conditions and molecular biology.** *Mycobacterium smegmatis* mc²-155 (ATCC) was grown in Middlebrook 7H9 liquid broth supplemented with: 10% albumin-dextrose-salt (ADS), 0.2% glycerol and 0.05% Tween-80, or plated on Lysogeny Broth (LB, Lennox) agar. *Mycobacterium tuberculosis* H37Rv (Beijing Chest Hospital strain), or clinical isolates of *M. tuberculosis* or *Mycobacterium bovis*-BCG (ATCC) strain Pasteur 1173P2 (BCG) were grown in 7H9 liquid broth supplemented with 0.2% glycerol and 10% oleic acid, albumin, dextrose, catalase (OADC) enrichment, or plated on 7H10-agar with 0.2% glycerol and 10% OADC enrichment. *E. coli* DH5α (CW Biotech) was used for cloning, as well as for the rifampicin plate assay, and was cultured in liquid LB broth or on LB agar. Antibiotic concentrations for *M. smegmatis* were: 50 µg ml⁻¹ hygromycin, 20 µg ml⁻¹ zeocin, 25 µg ml⁻¹ kanamycin. Antibiotic concentrations

for *E. coli* were: 150 µg ml⁻¹ hygromycin, 50 µg ml⁻¹ zeocin, 50 µg ml⁻¹ kanamycin. All strains were grown at 37 °C.

All primers used in the study are in Supplementary Tables 1, 2, plasmids and vectors used in the study are listed in Supplementary Table 3.

**Cell wall labeling and flow cytometry.** Fluorescent cell wall labeling strategy was adopted from[10] with some modifications. In all, 1 ml exponentially growing *M. smegmatis* culture (OD$_{600nm}$ = 0.4) was washed with PBST (phosphate-buffered saline (PBS) supplemented with 0.05% Tween-80) and then stained with amine reactive Alexa Fluor™ 488 (carboxylic acid, succinimidyl ester (Thermo Fisher Scientific)) at final concentration 0.05 mg ml⁻¹. Staining was >99% efficient routinely, and therefore loss of staining reflected growth. The stained cells were washed with PBST once, and then transferred into culture media containing antibiotics of interest and kept in 37 °C with shaking until further experimentation. All culture tubes were double covered with foil to avoid photo-bleaching. Cell cultures could be examined directly with flow cytometry, or pelleted and fixed in 2% paraformaldehyde (PFA) for up to an hour at room temperature or O/N at 4 °C. Cell wall fluorescence of fixed samples was found stable for up to a week when kept at 4 °C and in the dark. Stained and fluorescent (e.g., the strains expressing fluorescently tagged RpoB) samples were analyzed with BD Accuri™ C6 desktop flow cytometer (laser excitation: 488 nm, emission filter: 533/30 nm) and using Flow-Jo™ software. Antibiotic addition alone did not alter the fluorescence of unlabeled bacteria within the green channel.

**Rifampicin plate assay.** Exponentially growing bacteria cultures (optical density, OD$_{600nm}$ = 0.4 for *M. smegmatis* and *M. tuberculosis*, OD$_{600nm}$ = 0.6 for *E. coli*) were pelleted and re-suspended with culture media. Aliquots from multiple 10-fold dilutions were streaked on solid media (LB agar for *M. smegmatis* and *E.coli*, 7H10-agar with OADC enrichment for *M. tuberculosis*) containing corresponding concentration(s) of rifampicin with either sterile glass beads or disposable spreaders. Fraction survival was calculated as colony-forming units (CFUs) on rifampicin containing plates divided by the CFUs of non-selective agar (taking into account the dilution factor, as appropriate). Plate MIC was calculated as the concentration of rifampicin resulting in approximately 10% surviving number of colonies compared with non-selective media.

**Ethics statement.** Sputum samples were obtained from the Beijing City, Chaoyang District Tuberculosis Public Health Clinic. Samples were de-identified prior to analysis with a coded identifier, which was not shared with the research team at any time, but which would allow the clinical microbiologists to identify the source patient for follow-up sputum samples. Samples were from discarded sputum derived from patients as part of their routine clinical investigation and treatment, and not specifically sought as part of a study, and therefore the local IRB felt that no specific consent was required for the samples (15/5/17).

**Rifampicin plating assay of clinical *M. tuberculosis*.** Discarded sputum that was identified as sputum smear test positive by routine clinical investigation was collected from patients with no prior history of tuberculosis (i.e., suspected treatment naive). Subsequent standard phenotypic drug testing of the initial isolates as part of routine clinical investigation confirmed samples were fully drug susceptible. Outcome data with regard to potential future drug resistance in the isolates was not available. Sputum was decontaminated as per routine. Briefly, 2 ml sputum was mixed with equal volume 4% sterile NaOH and incubated for 15 min at 37 °C with gentle shaking. The mixture was then centrifuged at 3800 *g*, 3.5 ml of supernatant were aspirated and the pellet resuspended in 10 ml ice-cold PBS. The washing step was repeated two more times after which the pellet was resuspended in 1 ml ice-cold PBS. Aliquots were then taken, and 10 -fold dilutions plated onto supplemented 7H10-agar to calculate CFU, as well as 7H10-agar with 0.1 µg ml⁻¹ and 0.2 µg ml⁻¹ rifampicin for rifampicin plating phenotypic resistance as above. Plate CFUs were counted after 1 month.

**MIC determination.** The MIC$_{90}$ for plating was defined as the concentration of antibiotics in agar medium that resulted in 10% fractional survival. The broth MIC was determined by culturing 10⁵ CFU bacteria in 1 ml broth and antibiotics, with a no antibiotic control, in duplicates. The MIC was the antibiotic concentration at which there was no visible growth at the time when the no antibiotic control tube was turbid. It should be noted that MIC under broth, microfluidic and plate growth conditions varied considerably, and the appropriate concentrations were used throughout.

**Bactericidal activity (kill curve) analysis.** *M. smegmatis* cultures (OD = 0.6) were treated with either 1 µg ml⁻¹ rifampicin or dimethylsulfoxide (DMSO) for 3 h, then washed extensively with regular culture media to reduce drug carried-over. The washed samples were transferred into non-selective culture media, or media supplemented with rifampicin (10 or 50 µg ml⁻¹), isoniazid (50 µg ml⁻¹), streptomycin (1.25 µg ml⁻¹) or ofloxacin (1.25 µg ml⁻¹). In all, 100 µl culture aliquots were sampled immediately, or 8, 18, 32, 48 h after inoculation, and subjected

to serial 10-fold dilutions. In total, 10 µl were taken from each dilutions and spread onto plain LB agar to measure the survival CFUs.

**Flow cytometry sorting.** Fluorescence-activated cell sorting (FACS) was carried out using a BD FACSAria special order research product. Approximately $10^8$ AF488-stained *M. smegmatis* cells were inoculated into 7H9 media containing 0, 10, 100 µg ml$^{-1}$ rifampicin, and cultured at 37 °C for 18 h. From the 10 µg ml$^{-1}$ rifampicin-treated culture, one million cells of each gate in Fig. 2a were sorted into sterile falcon tubes. Meanwhile, one million cells from either 100 µg ml$^{-1}$ rifampicin-treated or -untreated cultures were also collected with identical settings. Sorted cells were spun down and re-suspended with fresh 7H9 media. Aliquots of multiple 10-fold serial dilutions of the cell suspensions were plated on antibiotic-free LB agar in triplicates. CFUs were enumerated after 5 days incubation at 37 °C to allow full recovery of survivors, and percentage survival rate was calculated.

**Fluorescence microscopy.** Steady fluorescent imaging data for Fig. 1a, Supplementary Figure 7b and 8 were acquired by inverted DeltaVision Elite widefield fluorescence microscope (GE Healthcare Life Sciences). Bacterial samples fixed with 2% PFA were spotted on 1.0% PBS agarose pad. DNA staining (Supplementary Figure 7b) was carried out by incubating fixed cells with 1 µg ml-1 Hoechst 33342 (Sigma) at room temperature for 10 min, and washed with PBS + 0.05% Tween-80 for three times to minimize dye carried-over. Steady fluorescent imaging data for Supplementary Figure 1 was acquired on a Nikon TI-E inverted microscope. Time lapse imaging was performed on a Nikon TI-E inverted microscope with an environmental chamber maintained at 37 °C. Mid-log-phase cultures were injected into a B04A microfluidic bacteria plate (CellASIC) to optimal density, then the cell chamber perfused with 7H9 medium. After 8 h, 7H9 medium supplemented with 10, 20, or 30 µg ml$^{-1}$ rifampicin was perfused into parallel flow chambers within the same chip and continued for 30 h.

**Image analysis.** Fluorescent image analysis was carried out with Fiji[59]. Background fluorescent signals were subtracted with rolling ball method. For each cell in the time series, cell axis was defined by the segmented line tool, and was used as the region of interest to extract its cell length, mean fluorescent intensity, as well as fluorescent intensity per pixel along the cell length. For Supplementary Fig. 11a, to quantify the lag time of RpoB accumulation or growth resumption in 33 cells, the moving average of cell lengths and mean fluorescent intensities in time series were calculated with window size equal to 1 h (four consecutive time points). From the moving averaged datasets, peak time was defined as the first point from which cell length or mean fluorescence intensity increased continuously over 1 h, lag time was hence the interval time between rifampicin perfusion to peak time of each cell. For Supplementary Fig. 11b, instead of the segmented line tool, the average intensity of three-point measurements in the nucleoid region of each cell was used to estimate RpoB intensity.

**rpoB-rpoC overexpression and analysis.** *M. smegmatis* was transformed with vector pUV15-tetOR::*rpoB-rpoC* or empty vector for the experiments. Exponentially growing bacteria at mid-log phase were induced with ATc at the indicated concentrations for 16 h. In all, 0.8 ml cultures were pelleted and frozen and used for western blotting (see below) and the remainder of the culture was stained with AF488 as above and then assayed ± rifampicin as above.

Western blotting was performed with antibodies (anti-DnaK, TS29, Abcam; anti-RpoB 8RB13, Santa Cruz, both used at 1:1000 dilution)[14].

**M. smegmatis RpoB-FP fluorescent reporter construction.** DNA sequence flanking the last 576-bp upstream of the stop codon of *rpoB* gene was amplified from *M. smegmatis* genomic DNA with primer pairs rpoB_FP_F and rpoB_mA_R (for RpoB-mApple construction) or rpoB_FP_F and rpoB_mEm_R (for RpoB-mEmerald construction). Similarly DNA sequence flanking the 486-bp downstream of the stop codon of *rpoB* gene was amplified with primer pairs ZeoR_rpoB_F and rpoB_FP_R. DNA sequence of the FPs mApple and mEmerald with C-terminal 6×histidine tag was amplified from pMV261-mApple or pMV261-mEmerald (a kind gift from the Rubin laboratory) with forward primer rpoB_mA_F or rpoB_mEm_F, and reverse primer rpoB_FP_R. The zeocin-resistant cassette flanked with *loxP* sites was amplified from pKM Zeo-lox plasmid using primer pairs FP_zeo_F and FP_zeo_R. Overlapping sequences were introduced into each fragment through PCR primers for further fusion with overlap. The final fusion product was purified and cloned into pCloneJet1.2 (Thermo Scientific) for sequence verification. Correct sequences were PCR amplified, purified, and electroporated into *M. smegmatis* expressing recombinase RecET[14]. *Bona fide* recombinants expressing RpoB-FP were verified directly with fluorescent microscopy and western blotting.

**M. smegmatis rpoB-rpoC Promoter-mEmerald construction.** Progressively truncated *rpoB-rpoC* promoter fragments were PCR amplified from *M. smegmatis* genomic DNA using forward primer set PrpoBC-F 1–19 and universal reverse primer PrpoBC-R. To make the Promoter I only construct, two fragments flanking the up- and downstream of Promoter II were amplified with primer pairs PrpoBC-

4-F/PrpoBC-20-R and PrpoBC-20-F/PrpoBC-R, and joined together with overlap PCR. The DNA sequence encoding mEmerald FP was amplified from pMV261-mEmerald using primer pair PrpoBC-mEm-F and *Hind*III-FP-R. The L5 integrating plasmid pML1342[60] was linearized with *Spe*I and *Hind*III double restriction digestion. Each *rpoB-rpoC* promoter variant was fused with mEmerald sequence by overlap PCR with universal primer pair *Spe*I_PrpoBC_F and *Hind*III_FP_R, column purified, digested with *Spe*I and *Hind*III, and then ligated into linearized pML1342 using QuickLigase (NEB).

The chimeric Psmyc-tetO-Promoter II-mEmerald constructs (P21 and P22) were constructed as below: the Psmyc-tetO region was amplified from pSE100 using primer pair Ptet-F/P21-tet-R (for P21 construct) or Ptet-F/P22-tet-R (for P22 construct). Corresponding Promoter II-mEmerald sequences were amplified from pML1342-P4-mEmerald plasmid using forward primer set Promoter II-21-F or Promoter II-22-F and universal reverse promoter UVtet-mEm-R. The vector pUV15tetOR[61] was linearized with *Pac*I and *Eco*Rv double restriction digestion, and the cognate two fragments of P21 or P22 were inserted into the linearized vector through Gibson assembly[62] and verified by Sanger sequencing.

**BCG rpoB-rpoC Promoter-mEmerald reporter construction.** The full-length *rpoB-rpoC* promoter was amplified from *M. tuberculosis*-H37Rv genomic DNA (the promoter sequences of BCG and *M. tuberculosis* are identical) with promoters TB_BCP_F and TB_BCP_R and fused to sequence of mEmerald as above. To create truncated promoters, the sequence upstream of P1 consensus was amplified with TB_BCP_F and TB_BCP_P1_del_R and the sequence downstream of P1 consensus was amplified with TB_BCP_P1_del_F and TB_BCP_F. The two fragments were fused together through overlap PCR using TB_BCP_F and TB_BCP_R, creating rpoB-rpoC Promoter depleted of P1 consensus. A similar approach was used to create a promoter lacking P2. The fused products were sub-cloned into pML1342 by restriction digestion/ligation through *Xba*I/*Spe*I sites. The plasmid was transformed to BCG using standard methodology and experiments measuring *rpoB-rpoC* expression using flow cytometry were performed as above.

**Reverse-transcriptase quantitative PCR.** Exponentially growing *M. smegmatis* or BCG were treated with either rifampicin at indicated concentrations or DMSO in duplicate prior to RNA extraction. RNA samples were extracted from each sample with method described in Su et al.[14]. Residual genomic DNA was removed by treating RNA samples twice with Turbo Dnase (Ambion). RNA concentrations were determined with Qubit$^{TM}$ (Life Technologies). Complementary DNA libraries of each sample were synthesized from 500 ng total RNA using iScript supermix (Bio-Rad). Quantitative PCR was carried out in triplicate with primers described in Supplementary Table 1 using iQ SYBR Green Super Mix (Bio-Rad). The $2^{-\Delta\Delta Ct}$ calculation was used to calculate the relative expression level of rifampicin-treated to -untreated samples (normalized to *sigA*).

**In vitro transcription.** RNA polymerase holoenzyme was prepared as described in Javid et al.[11]. Double-stranded DNA (dsDNA) encoding full-length *rpoB-rpoC* promoter and Promoter I only were amplified from pML1342-P1-mEmerald and pML1342-P20-mEmerald with primer pairs PrpoBC_F and PrpoBC_R. Promoter II DNA was amplified from pML1342-P1-mEmerald using primer pairs PrpoB-C_II_F and PrpoBC_R. The control dsDNA encoding Psmyc promoter sequence was amplified from pML1357[60] using primer pair Psmyc_F and Psmyc_R. Amplified dsDNA constructs were then column purified, dialyzed, and the final concentrations were adjusted to 1 pmol g$^{-1}$. The sequences of the templates used for the reaction are listed in Supplementary Table 4. In all, 20 µl reaction containing 200 ng purified RNAP holoenzyme, 1 pmol dsDNA template, ribonuclease inhibitor (Transgen), 10 mM Tris-HCl (pH = 8.0), 50 mM KCl, 10 mM MgCl$_2$, 2 mM DTT, 0.1 mM EDTA, 1 µg Nuclease-free BSA (Sigma), and rifampicin at indicated concentration was preheated to 37 °C and incubated for 15 min. In total, 5 µl NTPmix (ATP, CTP, UTP, GTP, 25 mM each) was then added to each reaction, briefly mixed by gentle vortex, and then incubated at 37 °C for 40 min. In all, 25 µl STOP solution (DnaseI (NEB), 2.5 µl 10 × DnaseI buffer (NEB), 0.15 µl Ribogreen$^{TM}$ nucleotide dye (Thermo), 0.5 µl rifampicin solution (1 mg ml$^{-1}$), add diethyl pyrocarbonate- (DEPC-) treated water to 25 µl) was added to each reaction, and template DNA was digested at 37 °C for 15 min. In total, 48 µl of each terminal reaction was transferred to black, flat bottom polystyrene 96-well plate, and fluorescence was measured by Fluoroskan Ascent luminometer with 100 ms integration time, excitation/emission filter at 488 nm/520 nm. Background signal was determined by three parallel reactions, which had their NTPmix substituted with DEPC-treated water. Each experiment was carried out in pentaplicate, the highest and the lowest readouts of all reads were excluded from final data analysis.

**M. smegmatis Promoter II-only strain construction.** The Promoter II-rpoB DNA fragment and *rpoC* DNA fragment were amplified from *M. smegmatis* genomic DNA with primer pairs 1357-Promoter II-rpoB-F/rpoB-rpoC-R and rpoB-rpoC-F/1357-rpoC-R. Giles site integrating plasmid pML1357[60] was linearized by *Spe*I single digestion and had its end phosphates removed with Antarctic phosphatase (NEB). The two fragments were inserted into pML1357 vector using Gibson assembly method and verified through Sanger sequencing. This integrating plasmid was then transformed into wild-type *M. smegmatis* mc²-155 to make the *rpoB-rpoC*

merodiploid strain. We then transformed the plasmid pNIT(kan)-RecET-SacB (kindly offered by the Rubin laboratory) into the *rpoB-rpoC* merodiploid strain, with which competent cells expressing RecET recombinases were prepared[14]. DNA sequences flanking the 500-bp upstream of the *rpoB-rpoC* promoter region and downstream of *rpoC* gene were amplified from genomic DNA using primer pairs rpoBC-KO-1/rpoBC-KO-2 and rpoBC-KO-5/rpoBC-KO-6. Zeocin-resistant marker flanked by loxP site was amplified from pKM Zeo-lox plasmid using primer pair rpoBC-KO-3 and rpoBC-KO-4. The three DNA fragments were then joined together using overlap PCR, and transformed into RecET expressing, merodiploid competent cells to knock out the native copy of *rpoB-rpoC* operon.

**Statistical analysis**. All experiments were performed at least three times on separate occasions except when noted otherwise. Data are presented as mean ± SD unless indicated otherwise. Means were compared by unpaired two-directional Student's *t*-test unless otherwise indicated.

## Data availability
The data that support the findings of this study are available from the corresponding author upon request.

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

## Acknowledgements

We would like to thank Eric Rubin for helpful discussions and suggestions. We would like to thank the staff at the Chaoyang District Tuberculosis Public Health Clinic for assistance in the experiments involving patient-derived sputa. This study was in part funded by grants from the Bill and Melinda Gates Foundation to BJ (OPP1109789) and also to J.-H.Z., start-up funds from Tsinghua University and grant 31570129 from the National Natural Science Foundation of China to B.J. B.J. is an Investigator of the Wellcome Trust (207487/B/17/Z).

## Author contributions

J.-H.Z. and B.J. designed the majority of the study and analyzed data, with some contributions from B.-W.W. J.-H.Z., B.-W.W., M.M.P. and Y.-N.Z. performed research. H.R. contributed reagents and contributed to the writing of the manuscript. J.-H.Z. and B.J. wrote the paper with input from the other authors.

## Additional information

**Competing interests:** The authors declare no competing interests.

