## [Peer Review File · Nature Communications]

Reviewers' comments:

Reviewer #1 (Remarks to the Author):

This is a highly original and significant contribution. It demonstrates a new mechanism for phenotypic tolerance to antibiotics, one of several that does not conform to the widely asserted presumption that non-replication, and hence non-dependence on biosynthetic processes, is a universal explanation for the phenomenon.

The paper traces this particular mechanism of rifampin tolerance to suppression of a weak, yet dominant, promoter for the gene encoding the target, RpoB, relieving the suppressive action of that promoter on a second, stronger one. The paper does not provide a specific molecular mechanism for the interference or its relief. However, discovery of further steps in the mechanism can appropriately be considered the subject for subsequent reports.

Major issues

1. The biggest problem with the paper is that the clinical relevance derives from its dealing with *M. tuberculosis*, but all the data shown pertain to *M. smegmatis* except for Fig. 1E and all that shows is a similar trend in CFU reduction as in *Msmeg* with higher concentrations of RIF. *M. smegmatis* is intrinsically resistant to rifampin relative to *M. tuberculosis*. It is not convincing that the major points of the study hold for *M. tuberculosis*. Key experiments need to be added that are conducted in *M. tuberculosis*. For example, Is the distribution of growing (dim) vs non-growing cells or the % of cells with increased RpoB expression identical or completely different in *Mtb* vs *Msmeg*? In the clinic RIF is co-administered to TB patients along with three other drugs. Did the authors test the impact of a combination treatment on the phenotypes observed?

2. The paper does not give a clear message whether increased expression of RpoB does or does not explain tolerance to rifampin. Lines 191-194 argue that RpoB abundance prior to rifampin exposure did not correlate with survival during rifampin exposure, whereas lines 211-213 argue that rifampin tolerance depends on RpoB accumulation.

3. The tolerance phenotype after RIF exposure caused by heterogeneous RpoB expression could be a general response to RNA polymerase perturbation. Have the authors tested other known, distinct classes of RNAP inhibitors such as fidaxomicin, myxopyronin, AA-phenylalaninamides etc., for their impact on RpoB expression and more broadly on their ability to induce tolerance? These compounds have been shown to inhibit *Mtb* survival. The term 'rifampicin-specific phenotypic resistance' could be a misnomer if other RNAP inhibitors cause a similar phenotype.

4. The authors use a clever adaptation of the CFSE labeling technique used to measure immune cell proliferation to monitor mycobacterial division. What is missing is the exact method used to determine the signal intensity cut-offs to distinguish growing and non-growing cells. There is a reference to unstained cells in Figure 1c, should this not be the vehicle control? The more accurate controls would be a comparison of live (vehicle control) and dead cells (obtained by heat-killing, for example) and then using these intensity distributions to determine the cut-off for 'growing' cells. The unstained control should be used for each treatment arm, since drug exposure by itself is known to alter auto fluorescence of mycobacteria in the GFP range. Also, the % of growing cells (dim cells) at MIC₉₀ of 10 ug/mL is approximately between 40-50% (Fig 1b). Does this mean that most of the dim cells grow initially, but then stop growing and thus do not form colonies? Isn't four-fold difference between the number of cells recovered by CFU (10% survival; Fig 1d) and those detected as tolerant by the staining (40-50%; Fig 1b) significant?

5. A 2-fold increase in RIF-MIC reported in supplementary table 1 is used to support the claim that pre-exposure to RIF leads to increased tolerance, however in most standard MIC assays a 2-fold shift is considered to be within the errors of the dilution method used. It would be useful to include

the statistical analysis of this data to know how significant the 2-fold shift is.

6. It is interesting that there is heterogeneity even within the RpoB over-expressing subpopulation observed after RIF exposure. However, it is less clear what this means. Did the authors observe asymmetric distribution of RpoB within cells of Type IV class versus Type V class? It would be useful to verify the semi-heritable nature of the tolerance phenotype at the level of RpoB expression. Does expression levels of RpoB normalize to vehicle control when fresh medium is added for a short time after RIF exposure to explain Figure 1f? Does a subsequent RIF exposure give the cells belonging to Types III-V any survival advantage? Also, if only 13% of total cells showed increased RpoB expression (Fig 3e), it is difficult to understand how the mean fluorescence intensity values (Fig 3b) increased by 50%. Was RpoB intensity many-fold higher in the small fraction of over-expressing cells?

7. Do the authors have any mechanistic understanding or speculation of why RpoB expression from Promoter-II is not inhibited by RIF?

8. Some of the data presentation is vague. For example, in page 5 the authors describe how mutations in RpoB cannot explain the high proportion of surviving colonies observed after plating on RIF-plates. However, there is no information on how these '24/24' colonies were isolated or what concentration RIF was used (25 or 100 ug/mL). Later in the text, colonies picked from a RIF-25 ug/mL are tested for phenotypic tolerance and a 10-fold increase is mentioned. The authors assume that none of the clones carried RpoB mutations, but during each experimental set up the initial inoculum is different genetically. It is quite possible there are pre-existing RpoB mutant clones, which are genetically RIF-resistant and contribute to the 10-fold increase in tolerance. In figures 1c and 2a, x and y-axis legends are missing. Such information might be useful for investigators trying to use this new protocol/assay. It would also be useful to include what % of the total number of sorted cells belonged to each of the different sub-fractions mentioned in Figure 2a.

9. The experiments are all conducted at least three times but many of the quantitative results appear to come from biologic triplicates in a single experiment. It is important to present aggregate results from all independently conducted experiments of each type that are subject to quantitation.

Minor issues:

1. The clinical recommendation to use higher doses of rifampin (lines 383-392) is well taken. The authors might consider additionally endorsing the concept of therapeutic drug monitoring. Peak rifampin blood levels at the standard dose vary about 100-fold in different individuals, and some 40% or more of TB patients receiving the drug are under-treated. Finger-prick blood-spots can now be obtained, stored at room temperature and shipped to a center with LC-MS capacity to determine rifampin levels and adjust them accordingly.

2. In Fig 2b, the authors have % live cells on the y-axis with a scale from 0 to 1.2. There is clear discrepancy between the text describing the results, which mentions 100% of the 7H9 control survived, and the graph that indicates 1%. Please correct the error.

3. Co-localization data reported in figure 4b is missing quantitative information. The supplemental figure 13a also does not give any clear numerical value.

4. Lines 158-165: The reporter switches from Emerald to Apple within the same experiment—this is confusing.

5. There are a number of run-on sentences, including one with consecutive "however's" (lines 49-50).

Reviewer #2 (Remarks to the Author):

Shu et al. describe important work showing that rifampicin tolerance in mycobacteria is, in part, controlled by an unusual self-regulatory loop on RNAP. This work is an important step forward for the TB field and more generally, as it molecularly describes a mechanism for drug tolerance. I believe that this work will be read with interest by the Nature Communications readership after some modifications and clarifications to the writing, as outlined below:

- 1) This regulatory system is an elegant modification of a type 4 incoherent feed, I believe. It may be more readable to some of the readership to describe it as such.
- 2) I believe that the abstract should be revised. The abstract begins too broadly and sometimes verges on overstating conclusions, for example suggesting strategies to "speed-up tuberculosis treatment". Another minor point here is that the authors claim that it is surprising that rifampicin exposure upregulates *rpob*, yet later cite work on the same claim.
- 3) An important paper from one of the coauthors, Rego et al 2017 has been published presumably after the submission of this work and it should be cited here.
- 4) MIC levels in constant flow are quite different from static/bulk culturing conditions and plating conditions. The authors should be careful to not suggest that they are equivalent. Because of this different, more clarity on how to compare numbers across the paper would be helpful.
- 5) It would be helpful and more convincing to see sample images of the staining plus brightfield imaging corresponding to figure 1C beyond the single image shown in the supplementary materials.
- 6) I am confused how to interpret Figure 2b but perhaps I simply do not understand the legend. I believe the left bar (tan) are cells that were not treated with rifampicin. So why is there only 1% live cells?
- 7) Figure 2c - I am unable to understand the x-axis label "hours post inoculation" or the details of this experiment. Does the time refer to treatment length?
- 8) Figure 4g - the figure for P21 and P22 look exactly the same on my print out.
- 9) Figure 4e is an important figure but is difficult to follow. Perhaps the authors could expand on the representative bar below the x-axis to make it more intuitive to follow how the different strains are related to the operon structure.
- 10) The statistics are largely based on a t-test. These statistics would be more convincing if data showing that there are underlying normal distributions are displayed.
- 11) line 164 - "lethal concentration" may not be the best word choice given that some bacteria survive and apparently thrive at this level.
- 12) I had to reread the section starting on line 215 many times and draw out my own diagrams to follow these data. I wonder if cloning details could be moved to methods, leaving more space (both in text and figures) to make these elegant experiments easier to process on the first read.

13) The comparison of cell size to reference 21 in the discussion is misleading. In both that study and Rego noted above, cell size was different in rifampicin tolerant and susceptible cells, but in both these studies tolerance was scored based on regrowth in the absence of drug. This is a different type of drug tolerance (leading to very different levels of "tolerance" by each scoring method), making these studies difficult to compare to the data presented here.

14) Fluorescent images that are not merged are better displayed in grayscale with pseudocolor reserved for the merged image.

15) Superfluous colors (for example in the bars in Figure 1) should be removed, especially if the color gradient is not consistent across subfigures. It is confusing to the reader.

Reviewer #3 (Remarks to the Author):

In this manuscript, Zhu et al. elucidate a mechanism of phenotypic rifampin resistance in *Mycobacterium smegmatis*. The authors demonstrate that there is heterogeneity in the mycobacterial response to rifampin where some cells are able to survive and grow in the face of rifampin treatment (at an MIC90) because they upregulate *rpoB* gene expression through the effects of a two promoter regulatory system.

Strengths of the manuscript are the novel and interesting insights into mycobacterial survival in the face of drug, the nice microscopy and the mechanistic parsing of the *rpoBC* promoter architecture. The weakness is that the authors have captured a sort of intermediate step in the overall process by which the cells that survive RIF emerge. It remains unclear why *rpoBC* expression is upregulated in some cells (where I think RIF should at least attenuate if not fully inhibit transcription even in the face of mistranslation induced RpoB mutations). It is also unclear why increased RpoB expression durably allows bacterial growth –here I agree with the authors that there must be associated changes in pump expression or genes that alter cell wall composition.

->The evidence of heterogeneous phenotypic RIF resistance is strong and convincing, using a nice combination of dynamic imaging and more conventional single point assays. Similarly, the dissection of the regulatory interactions between the two promoters is well done.

->Upregulation of RpoB expression in the setting of RIF exposure (as assessed by fluorescent protein reporters/fusions) is surprising and mechanistically not explained. Don't the authors have the strains to assess the mistranslation hypothesis they put forth (hyper-mistranslation should be associated with more cells that upregulate *rpoBC* expression after RIF exposure)?

Link between modest increase in RpoB as assessed by MFI and durable survival unclear especially because correlation doesn't hold for RpoB levels pre-treatment and RIF survival. Is it possible to sort the survivors (based on one of those reporters) and transcriptionally profile to identify other gene expression changes that might explain phenotype? (I admit that there is not a great control here and thus, I don't think this is required to complete this story.)

Minor

This is a data rich manuscript and would be easier to follow if it were trimmed (especially the supplementary data). One good place to cut might be the description of the 5 phenotypically defined subpopulations that emerge after drug treatment. This doesn't add much as these populations don't inform later analyses.

Reviewer #1

We thank the Reviewer for taking the time to make the following detailed observations and comments regarding our manuscript, and in particular for mentioning that our work is both “highly original and significant”.

1. The biggest problem with the paper is that the clinical relevance derives from its dealing with M. tuberculosis, but all the data shown pertain to M. smegmatis except for Fig. 1E and all that shows is a similar trend in CFU reduction as in Msmeg with higher concentrations of RIF. M. smegmatis is intrinsically resistant to rifampin relative to M. tuberculosis. It is not convincing that the major points of the study hold for M. tuberculosis. Key experiments need to be added that are conducted in M. tuberculosis. For example, Is the distribution of growing (dim) vs non-growing cells or the % of cells with increased RpoB expression identical or completely different in Mtb vs Msmeg? In the clinic RIF is co-administered to TB patients along with three other drugs. Did the authors test the impact of a combination treatment on the phenotypes observed?

Thank you for raising these important points. We have performed a number of additional experiments to specifically address the relevance of our observations for *M. tuberculosis*. Specifically, with regards to conservation of the rifampicin tolerance, we now show that pre-exposure to rifampicin increases rifampicin tolerance in the lab-adapted strain *M. tuberculosis*-H37Rv (Fig. 1f). Importantly, *Mtb* isolated directly from sputum patients with rifampicin-sensitive tuberculosis shows increased rifampicin plate-tolerance over time (and standard therapy drug exposure) – Fig. 1h. This experiment also critically addresses the issue of whether the rifampicin tolerance phenotype is dominant: since the patients were on standard therapy (i.e. rifampicin, isoniazid, pyrazinamide and ethambutol). As a further

additional set of experiments addressing this point (in *M. smegmatis*), we have shown the rifampicin-dominant effect in upregulation of RpoB (Supplementary Fig. 16). We were unable to perform experiments involving fluorescent microscopy or flow cytometry in our BSL-3 facilities due to lack of appropriate equipment in the BSL-3, but the vaccine strain *M. bovis*-BCG (BCG), a member of the *M. tuberculosis* complex has an identical *rpoB-rpoC* operon to *M. tuberculosis sensu stricto*. We therefore performed additional experiments mapping the promoter response to rifampicin in BCG. These show almost identical responses to *M. smegmatis* (Supplementary Fig. 18). Therefore both the general phenotype (rifampicin exposure induces rifampicin hypertolerance), and the behaviour of the operon: rifampicin exposure increases *rpoB* expression is identical between Msm and Mtb complex.

2. *The paper does not give a clear message whether increased expression of RpoB does or does not explain tolerance to rifampin. Lines 191-194 argue that RpoB abundance prior to rifampin exposure did not correlate with survival during rifampin exposure, whereas lines 211-213 argue that rifampin tolerance depends on RpoB accumulation.*

Thank you for raising this important point. Our data, including additional experiments suggest that the RpoB abundance within the cell, as measured by fluorescent-tagging of the native *rpoB* gene prior to high level rifampicin exposure are **not** predictive of future survival to rifampicin (Supplementary Fig. 12). However, upregulation of *rpoB* is clearly a conserved response in surviving and growing rifampicin-tolerant mycobacteria. Further experiments with supra-physiological over-expression of *rpoB-rpoC* from a tetracycline-inducible promoter **does** confer a survival advantage (Fig. 2e-g). These data suggest that physiological variation in RpoB abundance in the absence of RIF (which, the data in Supplementary Fig. 9 suggests is not very great) probably does not have a role in survival to rifampicin, but upregulation in response to RNAP inhibition (e.g. by rifampicin), or by expression *in trans*, does lead to survival. We cannot at this time resolve unequivocally whether this is due to abundance of rifampicin's target *per se*, or whether upregulation in response to stress initiates a specific survival programme, although we favour the latter hypothesis and we have commented on this in the Discussion.

3. *The tolerance phenotype after RIF exposure caused by heterogeneous RpoB expression could be a general response to RNA polymerase perturbation. Have the authors tested other known, distinct classes of RNAP inhibitors such as fidaxomicin, myxopyronin, AA-phenylalaninamides etc., for their impact on RpoB expression and more broadly on their ability to induce tolerance? These compounds have been shown to inhibit Mtb survival. The term 'rifampicin-specific phenotypic resistance' could be a misnomer if other RNAP inhibitors cause a similar phenotype.*

Thank you for the suggestion, and we have indeed now confirmed that the trigger for the response is RNAP inhibition, since e.g. fidaxomicin shows the same phenotype (Supplementary Fig. 6e and Supplementary Fig. 16e). We have therefore renamed the tolerance phenotype as RNAP-specific phenotypic resistance (RSPR).

4. The authors use a clever adaptation of the CFSE labeling technique used to measure immune cell proliferation to monitor mycobacterial division. What is missing is the exact method used to determine the signal intensity cut-offs to distinguish growing and non-growing cells. There is a reference to unstained cells in Figure 1c, should this not be the vehicle control? The more accurate controls would be a comparison of live (vehicle control) and dead cells (obtained by heat-killing, for example) and then using these intensity distributions to determine the cut-off for 'growing' cells. The unstained control should be used for each treatment arm, since drug exposure by itself is known to alter auto fluorescence of mycobacteria in the GFP range. Also, the % of growing cells (dim cells) at MIC90 of 10 ug/mL is approximately between 40-50% (Fig 1b). Does this mean that most of the dim cells grow initially, but then stop growing and thus do not form colonies? Isn't four-fold difference between the number of cells recovered by CFU (10% survival; Fig 1d) and those detected as tolerant by the staining (40-50%; Fig 1b) significant?

Thank you. We have added DMSO control as suggested (Fig. 1c). With regards to inclusion of heat-killed controls, despite repeated attempts, labelling of heat-killed bacteria with AF488 did not have high efficiency (not shown), leading to a significant proportion of unlabelled cells, which were undistinguishable in the histogram from growing cells, thus we have not included this condition in the final figure. The very small proportion of non-stained cells (0.4%) when cells were grown in very high concentrations of rifampicin (200µg/ml) we feel are validation that staining was >99.5% efficient, and we have added a note in the Methods to reflect this. We have also verified that antibiotic treatment does not alter fluorescence in the green channel (Fig. A, below, but otherwise not shown, except for a note in the Methods).

Fig. A. Antibiotic treatment does not alter green (FL1) fluorescence of *M. smegmatis*. Wild-type *M. smegmatis* was treated with antibiotics shown or DMSO (ND, no drug) for 4 hours and then fluorescence measured by flow cytometry in FL-1 channel.

With regards to the second point, regarding the difference in the size of growing tolerant population on plates (which represent 20 surviving divisions to form visible colonies) and loss of fluorescence overnight (4-6 divisions), sorting and plating of the non-fluorescent population onto non-selective medium (Fig. 1i), as well as the lineage study by microfluidics (Supplementary Fig. 10, 13) verifies that survival for one or two divisions in bulk-lethal selection conditions does not guarantee survival, and is the likely explanation for the disparity between the survival percentages under similar selection. Furthermore, viable non-fluorescent bacteria will only produce non-fluorescent progeny, even if those are subsequently killed, as verified in Fig. 1j, not all non-fluorescent bacteria were viable. We

have included the following wording: **“Furthermore, our data indicated that phenotypically resistant growers, or their progeny, were not guaranteed to survive rifampicin killing and that growers, persisters and killed cells co-existed upon rifampicin treatment.”**

5. A 2-fold increase in RIF-MIC reported in supplementary table 1 is used to support the claim that pre-exposure to RIF leads to increased tolerance, however in most standard MIC assays a 2-fold shift is considered to be within the errors of the dilution method used. It would be useful to include the statistical analysis of this data to know how significant the 2-fold shift is.

Thank you. We decided to revisit this question by examining antibiotic tolerance (with time-kill curves) to rifampicin, isoniazid, ofloxacin and streptomycin +/- pre-treatment with low-dose rifampicin (Supplementary Fig. 6), which we believe addresses the issue in a more unambiguous way. We now show clearly that pre-treatment with sub-MIC rifampicin clearly enhances rifampicin tolerance, but not tolerance to other antibiotics. These data support our hypothesis that RSPR is specific to agents targeting RNAP.

6. It is interesting that there is heterogeneity even within the RpoB over-expressing subpopulation observed after RIF exposure. However, it is less clear what this means. Did the authors observe asymmetric distribution of RpoB within cells of Type IV class versus Type V class? It would be useful to verify the semi-heritable nature of the tolerance phenotype at the level of RpoB expression. Does expression levels of RpoB normalize to vehicle control when fresh medium is added for a short time after RIF exposure to explain Figure 1f? Does a subsequent RIF exposure give the cells belonging to Types III-V any survival advantage? Also, if only 13% of total cells showed increased RpoB expression (Fig 3e), it is difficult to understand how the mean fluorescence intensity values (Fig 3b) increased by 50%. Was RpoB intensity many-fold higher in the small fraction of over-expressing cells?

Thank you for these comments. We re-examined our data for asymmetry of RpoB distribution (by measuring fluorescence of RpoB::mApple) and cell-size of type IV vs. type V cells (Supplementary Fig. 14) and cannot find differences to predict the differences in cell-fate. This may be due to the single time-point measurement of RpoB abundance, and we have addressed this in the Discussion.

With regards to resolving the apparent disparity between the old Figs. 3b and 3e, it should be noted that the referred to 3b (now 2b) was performed with sub-inhibitory rifampicin exposure, and therefore almost all cells increased *rpoB* exposure under those conditions. The 3e (now Fig. 2h) experiment was performed with bulk-lethal concentration of rifampicin, whereby most cells did not grow/ were killed, and only the subpopulation of cells with *rpoB* over-expression survived. We have clarified the difference in these conditions in the Figure legend.

7. Do the authors have any mechanistic understanding or speculation of why *RpoB* expression from Promoter-II is not inhibited by RIF?

Thank you. We believe that Promoter-II is relatively (compared with Promoter-I) resistant to inhibition by rifampicin. Higher concentrations of rifampicin did lead to inhibition of *in vitro* transcription from Promoter-II by RNAP (now shown in Supplementary Fig. 19b). Heterogeneity in promoter inhibition responses to rifampicin has been previously described (see e.g. PMID 23419780, cited in the Discussion).

8. Some of the data presentation is vague. For example, in page 5 the authors describe how mutations in *RpoB* cannot explain the high proportion of surviving colonies observed after plating on RIF-plates. However, there is no information on how these '24/24' colonies were isolated or what concentration RIF was used (25 or 100 ug/mL). Later in the text, colonies picked from a RIF-25 ug/mL are tested for phenotypic tolerance and a 10-fold increase is mentioned. The authors assume that none of the clones carried *RpoB* mutations, but during each experimental set up the initial inoculum is different genetically. It is quite possible there are pre-existing *RpoB* mutant clones, which are genetically RIF-resistant and contribute to the 10-fold increase in tolerance. In figures 1c and 2a, x and y-axis legends are missing. Such information might be useful for investigators trying to use this new protocol/assay. It would also be useful to include what % of the total number of sorted cells belonged to each of the different sub-fractions mentioned in Figure 2a.

Thank you for these comments. We have now clarified the origin of the colonies that were picked and sequenced and also tested a further 6 colonies from high-dose rifampicin agar (all wild-type sequence in *rpoB* RRDR). Furthermore, to clarify the issue regarding 10-fold increase in resistance, culture of the hyper-tolerant cells in the absence of rifampicin led to loss of the hyper-tolerance phenotype (Fig. 1g), excluding the possibility of heteroresistance. We have clarified this further in re-writing that element of the text: **“These observations could be consistent with “adaptive resistance”, which might suggest the phenomenon was mediated by mutations outside of the RRDR of *rpoB*. However, culturing bacteria in non-selective medium for 16 hours prior to plating onto rifampicin-agar led to a complete loss of the increased phenotypic resistance, suggesting that the “adaptive phenotypic resistance” was semi-heritable, and not mediated by genetic mutation (Fig. 1g).”** We have re-labelled the highlighted panels and axes, thank you for highlighting this for us.

9. The experiments are all conducted at least three times but many of the quantitative results appear to come from biologic triplicates in a single experiment. It is important to present aggregate results from all independently conducted experiments of each type that are subject to quantitation.

Thank you for raising this important point. We had aggregated data for most of the data and have done so for much of the data that we had not previously. For the flow cytometry

measurements of RpoB expression, small differences e.g. in drug exposure or timing led to substantial heterogeneity in **absolute but not relative** fluorescence. We have therefore kept the panels as representative data (which still include at least three biological replicates), for the sake of clarity. We do not believe this compromises the integrity of the data, and is in keeping with how similar data is presented elsewhere.

10. The clinical recommendation to use higher doses of rifampin (lines 383-392) is well taken. The authors might consider additionally endorsing the concept of therapeutic drug monitoring. Peak rifampin blood levels at the standard dose vary about 100-fold in different individuals, and some 40% or more of TB patients receiving the drug are under-treated. Finger-prick blood-spots can now be obtained, stored at room temperature and shipped to a center with LC-MS capacity to determine rifampin levels and adjust them accordingly.

Thank you for this comment, which we now further address in our concluding remarks.

11. In Fig 2b, the authors have % live cells on the y-axis with a scale from 0 to 1.2. There is clear discrepancy between the text describing the results, which mentions 100% of the 7H9 control survived, and the graph that indicates 1%. Please correct the error.

Thank you, corrected.

12. Co-localization data reported in figure 4b is missing quantitative information. The supplemental figure 13a also does not give any clear numerical value.

Thank you for this comment. Old Fig. 4b (now 3b) was not meant to show complete co-localisation, we apologise for the lack of clarity, and the same for old Supplementary Fig. 13a (now Supplementary Fig. 15). We used the two reporters to measure whether upregulation of β was transcriptional or post-transcriptional (translational), and this has been clarified in the text.

13. Lines 158-165: The reporter switches from Emerald to Apple within the same experiment—this is confusing.

Thank you for this comment. The choice of fluorescent protein was partly driven by availability of the constructs for the respective microfluidic and flow cytometry experiments at the time they were performed. We don't believe that choice of fluorescent protein fundamentally affects the validity of the results. We have revised the wording of this section to improve clarity.

14. There are a number of run-on sentences, including one with consecutive "however"s (lines 49-50).

Thank you, we have revised the text to improve the clarity of the writing.

Reviewer #2

We thank the reviewer for their positive comments regarding our manuscript, and in particular highlighting that the work is important for the TB field and more generally.

1. This regulatory system is an elegant modification of a type 4 incoherent feed, I believe. It may be more readable to some of the readership to describe it as such.

Thank you, we agree and have now commented on this in our discussion.

*2. I believe that the abstract should be revised. The abstract begins too broadly and sometimes verges on overstating conclusions, for example suggesting strategies to "speed-up tuberculosis treatment". Another minor point here is that the authors claim that it is surprising that rifampicin exposure upregulates *rpob*, yet later cite work on the same claim.*

Thank you for these comments. We have now revised the abstract, removing "Surprising", and also altering the final sentence to limit the conclusions to those drawn from the study itself.

3. An important paper from one of the coauthors, Rego et al 2017 has been published presumably after the submission of this work and it should be cited here.

Thank you, this study, and one from Sakatos *et al* 2018, showing a growth tolerance phenotype in *M. smegmatis* to isoniazid have now been cited.

4. MIC levels in constant flow are quite different from static/bulk culturing conditions and plating conditions. The authors should be careful to not suggest that they are equivalent. Because of this difference, more clarity on how to compare numbers across the paper would be helpful.

Thank you. We have reworded the methods to reflect that MICs under different conditions vary considerably.

5. It would be helpful and more convincing to see sample images of the staining plus brightfield imaging corresponding to figure 1C beyond the single image shown in the supplementary materials.

Thank you, this has now been performed (revised Supplementary Fig. 1)

6. I am confused how to interpret Figure 2b but perhaps I simply do not understand the legend. I believe the left bar (tan) are cells that were not treated with rifampicin. So why is there only 1% live cells?

Thank you for spotting this typo, which has now been corrected.

7. *Figure 2c - I am unable to understand the x-axis label "hours post inoculation" or the details of this experiment. Does the time refer to treatment length?*

Thank you and apologies for the lack of clarity. The panel describes the experiment now shown in Supplementary Fig. 5. We have clarified in the revised figure legend that “hours post inoculation” refer to time after picked colonies were transferred to the fresh medium +/- drug.

8. *Figure 4g - the figure for P21 and P22 look exactly the same on my print out.*

This has been corrected, thank you.

9. *Figure 4e is an important figure but is difficult to follow. Perhaps the authors could expand on the representative bar below the x-axis to make it more intuitive to follow how the different strains are related to the operon structure.*

Thank you for this comment. We have redrawn this figure (now 3d) with the axes swapped, to allow cartoons that better represent the truncations to be drawn alongside the axis.

10. *The statistics are largely based on a t-test. These statistics would be more convincing if data showing that there are underlying normal distributions are displayed.*

Thank you. We have now revised some of the statistical analysis, and used non-parametric tests where appropriate. This re-analysis had not changed the substantive findings of the manuscript.

11. *line 164 - "lethal concentration" may not be the best word choice given that some bacteria survive and apparently thrive at this level.*

Thank you. To improve clarity, we have now used the term “bulk-lethal” to refer to concentrations that kill >90% of the bacterial population.

12. *I had to reread the section starting on line 215 many times and draw out my own diagrams to follow these data. I wonder if cloning details could be moved to methods, leaving more space (both in text and figures) to make these elegant experiments easier to process on the first read.*

Thank you. We have now revised this section removing some of the cloning terminology to hopefully improve clarity.

13. *The comparison of cell size to reference 21 in the discussion is misleading. In both that study and Rego noted above, cell size was different in rifampicin tolerant and susceptible*

cells, but in both these studies tolerance was scored based on regrowth in the absence of drug. This is a different type of drug tolerance (leading to very different levels of "tolerance" by each scoring method), making these studies difficult to compare to the data presented here.

Thank you for this important comment, which we now address in the discussion.

14. Fluorescent images that are not merged are better displayed in grayscale with pseudocolor reserved for the merged image.

Thank you, now changed.

15. Superfluous colors (for example in the bars in Figure 1) should be removed, especially if the color gradient is not consistent across subfigures. It is confusing to the reader.

Thank you, now changed.

Reviewer #3

We thank the reviewer for their comments, and in particular in highlighting the novelty of the work.

1. Upregulation of RpoB expression in the setting of RIF exposure (as assessed by fluorescent protein reporters/fusions) is surprising and mechanistically not explained. Don't the authors have the strains to assess the mistranslation hypothesis they put forth (hyper-mistranslation should be associated with more cells that upregulate rpoBC expression after RIF exposure)?

Thank you. We have now used a high mistranslation strain from our previous study (Su, Zhu *et al* 2016) with a mutation in *gatA* to measure RSPR using our fluorescence dilution assay (Supplementary Fig. 6f, g). As predicted, the basal tolerance of this strain is higher than wild-type, and increases further upon low-dose rifampicin exposure. These revised data might explain the several logs increase in plating rifampicin tolerance seen with these strains in our previous studies.

2. Link between modest increase in RpoB as assessed by MFI and durable survival unclear especially because correlation doesn't hold for RpoB levels pre-treatment and RIF survival. Is it possible to sort the survivors (based on one of those reporters) and transcriptionally profile to identify other gene expression changes that might explain phenotype? (I admit that there is not a great control here and thus, I don't think this is required to complete this story.)

Thank you for this suggestion. We attempted to perform this experiment more than three times, but with no success. The major limitations were that we were unable to perform high quality RNAseq on fewer than 10^5 cells, even in log phase growth. Sorted cells were relatively "unhappy" and needed even more cells for quality RNA extraction and sequencing.

Sorting this many cells took several hours, and in the absence of ongoing rifampicin exposure – a necessary condition that was absolutely required during sorting – the phenotype of the sorted cells had partially reverted (as also seen in Fig. 1g), making interpretation of the data impossible.

3. This is a data rich manuscript and would be easier to follow if it were trimmed (especially the supplementary data). One good place to cut might be the description of the 5 phenotypically defined subpopulations that emerge after drug treatment. This doesn't add much as these populations don't inform later analyses.

Thank you for this comment. We have tried to streamline the paper by having 4 instead of 5 main data figures. However, almost all of the supplementary data (including specifically the subpopulation experiment) has been commented on by at least one reviewer, therefore we do not feel we can justify omitting these data. Additional experiments requested have also necessitated an increase in the number of supplemental panels.

REVIEWERS' COMMENTS:

Reviewer #1 (Remarks to the Author):

I re-reviewed the paper together with two labmates. I will present all three responses (A-C).

A. This is an excellent and important paper. I agree with the remaining concerns of reviewers B and C below. My advice is to avoid the generalization "mycobacteria(I)" throughout the text and legends and instead state the species used in every experiment and in every inference or conclusion where the data to justify the ascription are strong.

B. This contribution continues to be original and important to the field, and this revision is improved and responds to many of the reviewer's questions. The limitations of working with Mtb and patient samples are understandable, and the authors have attempted to perform some relevant experiments in Mtb, BCG, and in sputum samples from two patients. Subsequently, the conclusion of the Mtb portions of the paper are limited to an increase in tolerance to rifampicin with rifampicin exposure, but beyond this still requires inference from Msmeg. The BCG work has raised one particular concern regarding a key difference from Msmeg as described below that should be clarified prior to acceptance, even if full mechanistic elucidation of the issue can be reserved for future manuscripts.

Major issues:

1. The manuscript states on line 311 that in BCG, "as long as Promoter II was intact, rifampicin exposure consistently upregulated mEmerald." Indeed, supplementary figure 18 suggests continued *rpoB* induction by rifampicin despite the lack of promoter I in BCG. Doesn't this contradict in BCG the statement in line 320 that, "Promoter I was required for the rifampicin-induced expression of *rpoB*..."? Although the lack of promoter I clearly had an effect in increasing mEmerald expression in supplementary figure 18 without rifampicin, the subsequent proportional increase with rifampicin looks to be about equivalent to the increase with rifampicin exposure in the full length promoter. This is problematic for the later model in figure 5, since it is unknown for BCG if the rifampicin effect in vivo is due to inhibition of promoter I, the unexplained induction of promoter II with rifampicin independent of promoter I, or both. If BCG's promoter II behaves differently than Msmeg's with respect to reaction to rifampicin (i.e., independence from promoter I or the inter-promoter sequence), this should be discussed as it would suggest that Mtb might also behave differently than Msmeg in ways fundamental to the paper's conclusions.

2. Was there any evidence of rifampicin resistance among the sputum isolates after 1 and 2 weeks of therapy? It is unclear if there was subsequent testing after the initial testing. Conceivably a population genetically resistant to rifampicin could be enriched with regard to the remaining surviving Mtb in the patient even if it is not clinical practice to check so early on in treatment course. Overall, the language surrounding figure 1h in the text should be softened, as the results from sputa from two patients may be difficult to interpret given variability in clinical patients and samples.

Minor comments:

1. Especially in light of the issue above, it should be made clear in the text whether supplementary figure 19b is with BCG or Msmeg promoters.
2. Figure 1h shows a discrepancy with the text—was it 1 and 2 weeks following standard therapy initiation, or 1 and 3 weeks as the figure suggests?
3. Figure 4b appears to be mislabeled somehow—are the left bars of each set WT and the right bars the other strain?
4. Lines 56-60: Would break up this run-on sentence.
5. Line 306: "In" repeated twice

C. The authors have succeeded in providing more convincing and detailed data to describe this clinically relevant phenomenon of drug exposure induced RIF tolerance in mycobacteria. Most of the concerns raised by the reviewers have been addressed in this revised version of the manuscript and has helped in clarifying some of the mechanistic aspects of this phenotype. Future studies on the tentative role of other transcription factors and/or stress response pathways in this phenomenon, as discussed by the authors, will greatly benefit the TB community and help us gain understanding of mycobacterial response to RNAP inhibition.

Minor issues:

1. In order to address the role of RpoB promoter architecture in RSPR, a strain carrying only Promoter II upstream of RpoB has been used. As expected, the strain is inherently tolerant to RIF and has an almost identical or mildly higher fraction of surviving bacteria as WT after RIF pre-exposure (Fig. 4a&b). The authors conclude that the lower increase in survived fraction indicates lesser fine-tuning, however without any information on the expression level of RpoB in this strain (+/- RIF) it might be difficult to draw this conclusion. Is RpoB expression in the Promoter II-only strain increased after RIF pre-exposure? Based on Fig. 3d, m-Emerald fluorescence from the Promoter II-only constructs (P10-P13) does not increase after RIF pre-exposure, suggesting there are other downstream factors induced by RIF pre-exposure that work with the abundant RpoB molecule to induce this phenotype.

2. The authors have convincingly shown that pre-exposure to RNA polymerase inhibition stress induces a specific tolerance program in mycobacteria mainly mediated through RpoB up regulation. The phenotype appears to require less than 2-fold up regulation (Fig. 2b) of RpoB to manifest as growth or increase in length of tolerant cells (Fig. 2d), however a similar range of expression levels in unperturbed cells does not correlate with tolerance (Supplementary Fig. 12). Also, artificial up regulation of RpoB in a Tet-inducible strain with a dose as low as 1 ng/mL of ATC, which does not seem to lead to higher expression of RpoB compared to vector control (Fig. 2e lane 6 vs lane 2), does induce RSPR (Fig. 2g). Could this discrepancy be explained by some technical issues associated with the TetO strain? Otherwise it suggests there might be additional factors than just induction of late adaptive/stress response genes (lines 227-228) that is responsible for the RSPR phenotype. This should be addressed.

Reviewer #3 (Remarks to the Author):

The authors have adequately addressed my concerns and I think done a nice job of responding to the other reviews.

Response to Reviewer Comments

Reviewer #1

We thank the Reviewer and two of their colleagues for their detailed and constructive feedback, which has helped improve our article. In particular, we are grateful that all three agree that our work to be “excellent”, “important” and “original”.

With regards to specific comments raised:

Reviewer A

1. This is an excellent and important paper. I agree with the remaining concerns of reviewers B and C below. My advice is to avoid the generalization "mycobacteria(l)" throughout the text and legends and instead state the species used in every experiment and in every inference or conclusion where the data to justify the ascription are strong.

Thank you for this comment. We have gone through the manuscript, and in almost all cases removed the general terms “mycobacteria/ mycobacterial” and confirmed the species used in the experiment. A few general statements that applied to e.g. both Msm and Mtb (i.e. rifampicin exposure induced increased rifampicin tolerance) have retained the inclusive term “mycobacterial”, since experiments in both species showed the same phenomena.

Reviewer B

Major issues:

1. The manuscript states on line 311 that in BCG, “as long as Promoter II was intact, rifampicin exposure consistently upregulated mEmerald.” Indeed, supplementary figure 18 suggests continued rpoB induction by rifampicin despite the lack of promoter I in BCG. Doesn't this contradict in BCG the statement in line 320 that, “Promoter I was required for the rifampicin-induced expression of rpoB...”? Although the lack of promoter I clearly had an effect in increasing mEmerald expression in supplementary figure 18 without rifampicin, the subsequent proportional increase with rifampicin looks to be about equivalent to the increase with rifampicin exposure in the full length promoter. This is problematic for the later model in figure 5, since it is unknown for BCG if the rifampicin effect in vivo is due to inhibition of promoter I, the unexplained induction of promoter II with rifampicin independent of promoter I, or both. If BCG's promoter II behaves differently than Msmeg's with respect to reaction to rifampicin (i.e., independence from promoter I or the inter-promoter sequence), this should be discussed as it would suggest that Mtb might also behave differently than Msmeg in ways fundamental to the paper's conclusions.

Thank you for these observations. We have revised the wording in the Results to now specify which observations apply to which species, and where they diverge specifically. With regards to potentially different mechanisms for the role of Promoter I, the legend of the “model figure” (now Figure 6, since the legend of Figure 1 was too long, and that Figure has had to be revised to two figures), we have specifically mentioned in the caption that BCG/ Mtb may have a slightly different mechanism by which rifampicin exposure results in increased *rpoB* expression and rifampicin tolerance.

2. Was there any evidence of rifampicin resistance among the sputum isolates after 1 and 2 weeks of therapy? It is unclear if there was subsequent testing after the initial testing. Conceivably a population genetically resistant to rifampicin could be enriched with regard to the remaining surviving Mtb in the patient even if it is not clinical practice to check so early on in treatment course. Overall, the language surrounding figure 1h in the text should be softened, as the results from sputa from two patients may be difficult to interpret given variability in clinical patients and samples.

Thank you for these comments on our limited exploration of clinical Mtb and this phenotype. We have softened the wording with regards to extrapolation of our results: “...confirming this to be a potentially clinically relevant phenotype”, as well in the Discussion: “**Our preliminary and limited investigation of RSPR in two clinical isolates suggests that it may be clinically relevant.**”

With regards to phenotypic DST of the samples: these were performed on the originally isolated samples (pre-treatment) and reported to us as being rifampicin-sensitive. We are not aware of later testing of subsequent samples (which is not routine in the absence of treatment failure), nor the ultimate clinical outcome of the two patients, and we have reported this in the Methods.

Minor comments:

1. Especially in light of the issue above, it should be made clear in the text whether supplementary figure 19b is with BCG or Msmeg promoters.

We have now clarified in the text that it is the *M. smegmatis* promoter.

2. Figure 1h shows a discrepancy with the text—was it 1 and 2 weeks following standard therapy initiation, or 1 and 3 weeks as the figure suggests?

Thank you for spotting this typographical error, (it was 1 and 3 weeks), which has now been corrected.

3. Figure 4b appears to be mislabeled somehow—are the left bars of each set WT and the right bars the other strain?

Thank you for this comment. In the re-formatting of the panels for the revised

submission, this panel was mis-annotated in terms of shading, and this has now been corrected. The panel is now 5b.

4. Lines 56-60: Would break up this run-on sentence. AND Line 306: "In" repeated twice.

Thanks, amended and corrected respectively.

Reviewer C

Minor issues:

1. In order to address the role of RpoB promoter architecture in RSPR, a strain carrying only Promoter II upstream of RpoB has been used. As expected, the strain is inherently tolerant to RIF and has an almost identical or mildly higher fraction of surviving bacteria as WT after RIF pre-exposure (Fig. 4a&b). The authors conclude that the lower increase in survived fraction indicates lesser fine-tuning, however without any information on the expression level of RpoB in this strain (+/- RIF) it might be difficult to draw this conclusion. Is RpoB expression in the Promoter II-only strain increased after RIF pre-exposure? Based on Fig. 3d, m-Emerald fluorescence from the Promoter II-only constructs (P10-P13) does not increase after RIF pre-exposure, suggesting there are other downstream factors induced by RIF pre-exposure that work with the abundant RpoB molecule to induce this phenotype.

Thank you for these comments. We did not measure RpoB expression by Western in this strain. We agree that although the increase in rifampicin tolerance after rifampicin exposure is blunted compared with the wild-type strain, there is still an increase in response, which is not apparent in the studies of the promoter activity using the fluorescent reporters. We have therefore added the following wording to our findings: **“There was still a slight increase in rifampicin tolerance in this strain following rifampicin exposure, suggesting that either the new chromosomal location of the Promoter II-driven *rpoB-rpoC* locus, or other downstream factors may also play a role in tuning of rifampicin tolerance.”**

The authors have convincingly shown that pre-exposure to RNA polymerase inhibition stress induces a specific tolerance program in mycobacteria mainly mediated through RpoB up regulation. The phenotype appears to require less than 2-fold up regulation (Fig. 2b) of RpoB to manifest as growth or increase in length of tolerant cells (Fig. 2d), however a similar range of expression levels in unperturbed cells does not correlate with tolerance (Supplementary Fig. 12). Also, artificial up regulation of RpoB in a Tet-inducible strain with a dose as low as 1 ng/mL of ATC, which does not seem to lead to higher expression of RpoB compared to vector control (Fig. 2e lane 6 vs lane 2), does induce RSPR (Fig. 2g). Could this discrepancy be explained by some technical issues associated with the TetO strain? Otherwise it suggests there might be additional factors

than just induction of late adaptive/stress response genes (lines 227-228) that is responsible for the RSPR phenotype. This should be addressed.

Thank you for these comments. We agree that there is apparent discrepancy of the total bulk RpoB expression by Western, and rifampicin tolerance. Although we have no direct evidence; tolerance is, by definition, mediated by (usually a minor) subpopulation, and highly heterogeneous cell-cell variation in RpoB expression (and downstream responses) may mediate the phenotype, even when there is no observable change in bulk expression. We have commented on this in the Discussion as follows: **“Heterologous expression of RpoB-RpoC via a tetracycline-inducible plasmid led to increased tolerance, even at very low (1ng/ml ATc) levels of induction that did not result in observable increase in total RpoB by Western blot (Fig. 3f). This may be due to highly variable cell-cell variation in RpoB expression in a minor subpopulation mediating the phenotype, which would not be detected in the bulk population by Western blot.”**

Reviewer #3

The authors have adequately addressed my concerns and I think done a nice job of responding to the other reviews.

Thank you for these positive comments and assessment of our work.
